# Neighbourhood interactions drive overyielding in mixed-species tree communities

Andreas Fichtner [1], Werner Härdtle [1], Helge Bruelheide [2,3], Matthias Kunz [4], Ying Li [5] & Goddert von Oheimb [3,4]

Theory suggests that plant interactions at the neighbourhood scale play a fundamental role in regulating biodiversity–productivity relationships (BPRs) in tree communities. However, empirical evidence of this prediction is rare, as little is known about how neighbourhood interactions scale up to influence community BPRs. Here, using a biodiversity–ecosystem functioning experiment, we provide insights into processes underlying BPRs by demonstrating that diversity-mediated interactions among local neighbours are a strong regulator of productivity in species mixtures. Our results show that local neighbourhood interactions explain over half of the variation in observed community productivity along a diversity gradient. Overall, individual tree growth increased with neighbourhood species richness, leading to a positive BPR at the community scale. The importance of local-scale neighbourhood effects for regulating community productivity, however, distinctly increased with increasing community species richness. Preserving tree species diversity at the local neighbourhood scale, thus seems to be a promising way for promoting forest productivity.

[1] Institute of Ecology, Leuphana University of Lüneburg, Universitätsallee 1, 21335 Lüneburg, Germany. [2] Institute of Biology/Geobotany and Botanical Garden, Martin Luther University Halle-Wittenberg, Am Kirchtor 1, 06108 Halle (Saale), Germany. [3] German Centre of Integrative Biodiversity Research (iDiv), Halle-Jena-Leipzig, Deutscher Platz 5E, 04103 Leipzig, Germany. [4] Institute of General Ecology and Environmental Protection, Technische Universität Dresden, Pienner Straße 7, 01737 Tharandt, Germany. [5] School of Soil and Water Conservation, Beijing Forestry University, 35 Qinghua E Rd, Haidian District, 100083 Beijing, China. Correspondence and requests for materials should be addressed to A.F. (email: fichtner@leuphana.de) or to G.v.O. (email: Goddert_v_Oheimb@tu-dresden.de)

Tree species richness has been shown to foster ecosystem functions such as forest productivity[1–3], and biodiversity loss is expected to have negative implications for forest productivity worldwide[4]. Multiple studies in forests analysed biodiversity–productivity relationships (BPRs) at the scale of tree communities and found that tree species mixtures can yield higher productivity compared to monocultures (overyielding). Although positive BPRs were demonstrated by recent tree biodiversity experiments at both the community[5–8] and local neighbourhood scale[7,9,10], the mechanisms underlying BPRs are hardly understood. Specifically, it remained unclear, how tree interactions at the local neighbourhood level—the crucial scale of species interactions[11]—drive community BPRs. Exploring the way how individuals respond to changing neighbourhood conditions (e.g., neighbour diversity and abundance)[10,12,13] and how these neighbourhood interactions scale up to influence the community response, is therefore fundamental to understand the mechanisms underlying BPRs in tree communities[14].

The effect of species mixing on productivity (i.e., the net biodiversity effect) can result from multiple mechanisms, such as (1) selection effects, (2) resource partitioning, leading to competitive reduction, (3) facilitation and (4) natural enemy (e.g., pathogens or herbivores) partitioning, resulting in reduced Janzen–Connell effects (dilution effects)[15,16]. Statistically, the net biodiversity effect at the community scale can be partitioned in complementarity and selection effects[17]. While selection effects account for increased likelihood of including dominant and well-performing species in diverse communities, all other mechanisms of net biodiversity effects are summarised by the term 'complementarity'. Findings from tree biodiversity experiments provide support that positive BPRs result mostly from selection effects rather than complementarity effects[5]. However, there is also empirical evidence that tree mixtures enable higher canopy packing by means of niche differentiation in crown heights among species and intraspecific crown plasticity[18–21], which, in turn, contributes to increasing productivity of the community. Similarly, experimental and observational studies have shown that neighbourhood diversity increases individual tree growth through competitive reduction or facilitation[9,10,22–24]. Moreover, tree growth was found to be negatively related to damage of leaf fungal pathogens, which in turn decreased with tree species richness, thus showing a negative density dependence[25]. Such processes leading to overyielding in species mixtures can act at both the community[26] and neighbourhood[27] scale.

Mixed-species plant communities are the sum of co-occurring individuals of different species. As such, they can be considered as a network of locally interacting individuals[28]. Consequently, the response of tree communities to species mixing should be—at least to a certain extent—the result of aggregated small-scale variations in neighbourhood interactions[7,9,21,29]. Such neighbourhood interactions can either enhance or reduce individual tree growth, and are largely shaped by simultaneously operating positive (e.g., niche differentiation or facilitation) and negative (e.g., competition for resources) processes among neighbouring trees[30,31]. For example, simulation models revealed that neighbourhood interactions can induce positive BPRs in tree communities[24], but the extent to which locally interacting neighbours contribute to BPRs at the community scale is still poorly understood[32]. Specifically, empirical tests of the relationship between biodiversity effects at different spatial scales remain rare (but see ref. [33]), and the importance of neighbourhood interactions for enhancing productivity in mixed-species forests has not been quantified so far.

Here, we used tree communities of an early successional subtropical forest planted at two spatially explicit experimental sites —site A and B of a large-scale biodiversity–ecosystem functioning experiment in subtropical China (BEF-China)[34]—to quantify the contribution of neighbourhood interactions to biodiversity effects (using species richness as a measure for biodiversity) at the community scale (i.e., at the plot level). Our tree communities comprise 40 native broad-leaved species and cover a long diversity gradient, ranging from monocultures to 24-species mixtures. We hypothesise that positive BPRs in tree communities largely depend on how trees interact at the neighbourhood scale, and that the importance of neighbourhood interactions for BRPs increases as community species richness increases. To test these hypotheses, we applied a four-step approach: first, we used a neighbourhood modelling framework in which the annual wood volume growth (our measure for productivity) of a focal tree was expressed as a function of its initial size (wood volume) and the effects of neighbourhood competition (NCI), conspecific neighbour density (CND) and neighbourhood species richness (NSR). This analysis was based on 3962 focal trees growing at site A and allowed us to quantify individual-based biodiversity effects at the neighbourhood scale. In this study, we define the term 'individual-based biodiversity effect' as the net effect of all intra- and interspecific interactions within the neighbourhood of a focal tree (sensu ref. [17]), while neighbourhoods are defined as the total number of closest trees surrounding a focal tree with a maximum of eight neighbours (i.e., the local neighbourhood). Second, we predicted the annual wood volume growth of 3018 focal trees growing at site B, using parameter estimates obtained from the neighbourhood model of site A. Third, we calculated standardised plot-level aboveground wood productivity (AWP; hereafter community productivity) by summing size-standardised growth rates (separately for observed or predicted values) of all focal trees within a plot for site B. This allowed us to obtain and compare measures for observed ($AWP_{obs}$) and predicted community productivity ($AWP_{nbh}$), based on neighbourhood interactions. Finally, we applied a community-modelling framework in which $AWP_{obs}$ was expressed as a function of community species richness (CSR), $AWP_{nbh}$ and topography to account for variation in biotic and abiotic growing conditions. We then quantified the amount of variation in observed community productivity explained by neighbourhood interactions ($AWP_{nbh}$) along the diversity gradient, which allowed us to explore the link between biodiversity effects at the neighbourhood and community scale. Importantly, our function-derived growth rates were based on different data sets (site A data: neighbourhood model, site B data: community model) that represent different species pools (Supplementary Table 1), and thus ensure independence when examining the relationship between biodiversity effects at different spatial scales. Our study demonstrates that positive effects of biodiversity on community productivity are largely driven by interactions among local neighbours, highlighting the need to promote tree species diversity at the local neighbourhood scale for enhancing forest productivity.

## Results

**Biodiversity effects at the local neighbourhood scale.** Overall, we found positive effects of neighbourhood species richness (NSR) on individual tree growth ($G$), but the magnitude of biodiversity effects was determined by the focal trees' size (i.e., initial wood volume) and neighbourhood competition (NCI; Fig. 1 and Supplementary Table 2). Conspecific neighbour density was not significantly related to $G$ ($\chi^2$: 0.37, $P = 0.540$). Importantly, results from neighbourhood models fitted for focal trees growing at sites A and B, and based on different species sets, were qualitatively the same (Supplementary Table 3), suggesting that our estimates of $AWP_{nbh}$ had an adequate power to explore the link between $AWP_{obs}$ and $AWP_{nbh}$.

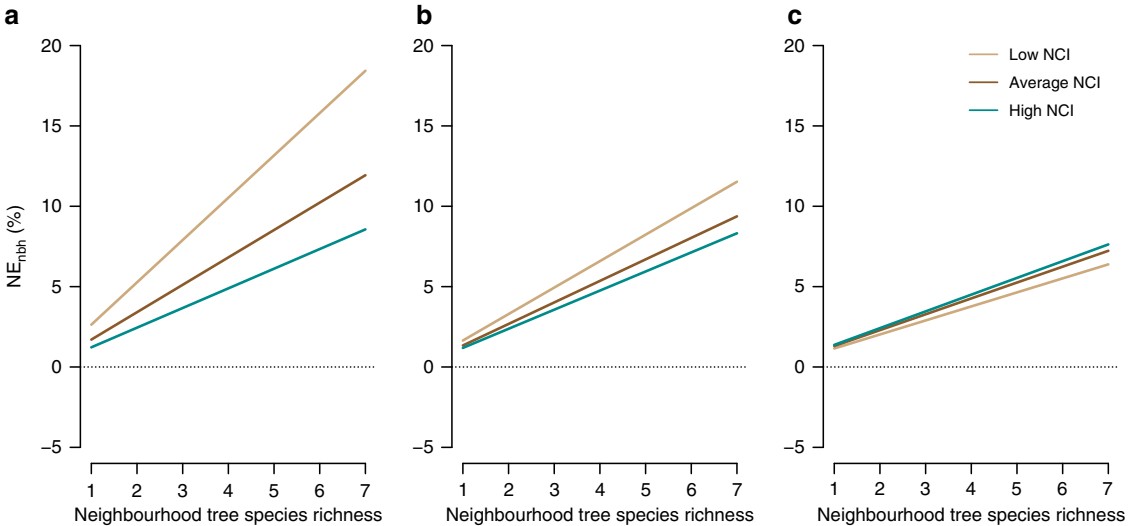

**Fig. 1** Biodiversity effects on individual tree growth. Size-dependent variation in net biodiversity effects at the neighbourhood scale ($NE_{nbh}$) with neighbourhood competition and neighbourhood tree species richness (NSR). $NE_{nbh}$ indicates the predicted change (%) in individual tree growth (annual growth rate of wood volume of a focal tree growing with heterospecific compared to growing with conspecific neighbours) in response to neighbourhood tree species richness at low, average and high value of neighbourhood competition index (NCI) for **a** small-sized, **b** medium-sized and **c** large-sized trees. Lines represent mixed-effects model fits for each size and competition level, respectively. Tree size, NCI and NSR explained 48% of the variation in individual tree growth

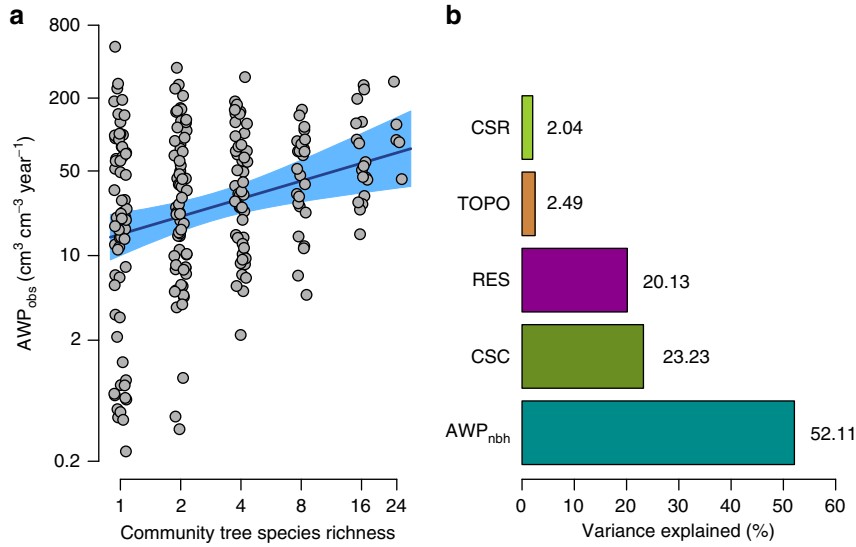

**Fig. 2** Biodiversity effects on community productivity. **a** Biodiversity–productivity relationship at the community scale. The solid blue line corresponds to the fitted relationship of a mixed-effects model, with the shaded area representing the 95% confidence interval of the prediction. Points represent observed values of standardised aboveground wood productivity ($AWP_{obs}$) for each individual plot ($n = 234$; site B). Plot-specific values are jittered to facilitate visibility, and axes are logarithmic. **b** Biotic and abiotic drivers of the community biodiversity–productivity relationship. Variance partitioning for four predictors: community (i.e., at the plot level) tree species richness (CSR) and composition (CSC, specified as random effect), community productivity based on neighbourhood inteactions ($AWP_{nbh}$) and heterogeneity in topography (i.e., variation in elevation, TOPO). Bars and numbers next to the bars correspond to the fraction of variance explained by each predictor of a linear mixed-effects model, and the variance not explained by the model (the residual, RES)

**Biodiversity effects at the community scale**. As expected, NSR was positively and strongly related to community species richness (CSR; $r^2 = 0.66$, $P < 0.001$; Supplementary Fig. 1), whereby positive neighbourhood-scale biodiversity effects translated into positive effects at the community scale. Consequently, observed community productivity increased with CSR ($AWP_{obs}$; $t = 3.25$, $P < 0.01$). On average, $AWP_{obs}$ of highly species-rich communities (24-species mixtures) was more than twice as high (+122%) as those of monocultures (Fig. 2a). $AWP_{obs}$ of monocultures was highly variable and was on average higher for species with low

wood density ($t = -3.08$, $P < 0.01$) and leaf toughness ($t = -5.58$, $P < 0.001$; Supplementary Fig. 2). Particularly, slow-growing species (i.e., species in the 25% quantile of species-specific $AWP_{obs}$ in monoculture) benefited the most from growing in species-rich communities (16-/24-species mixtures; Supplementary Figure 3).

**Importance of neighbourhood interactions**. The best-fitting community productivity model included positive effects of

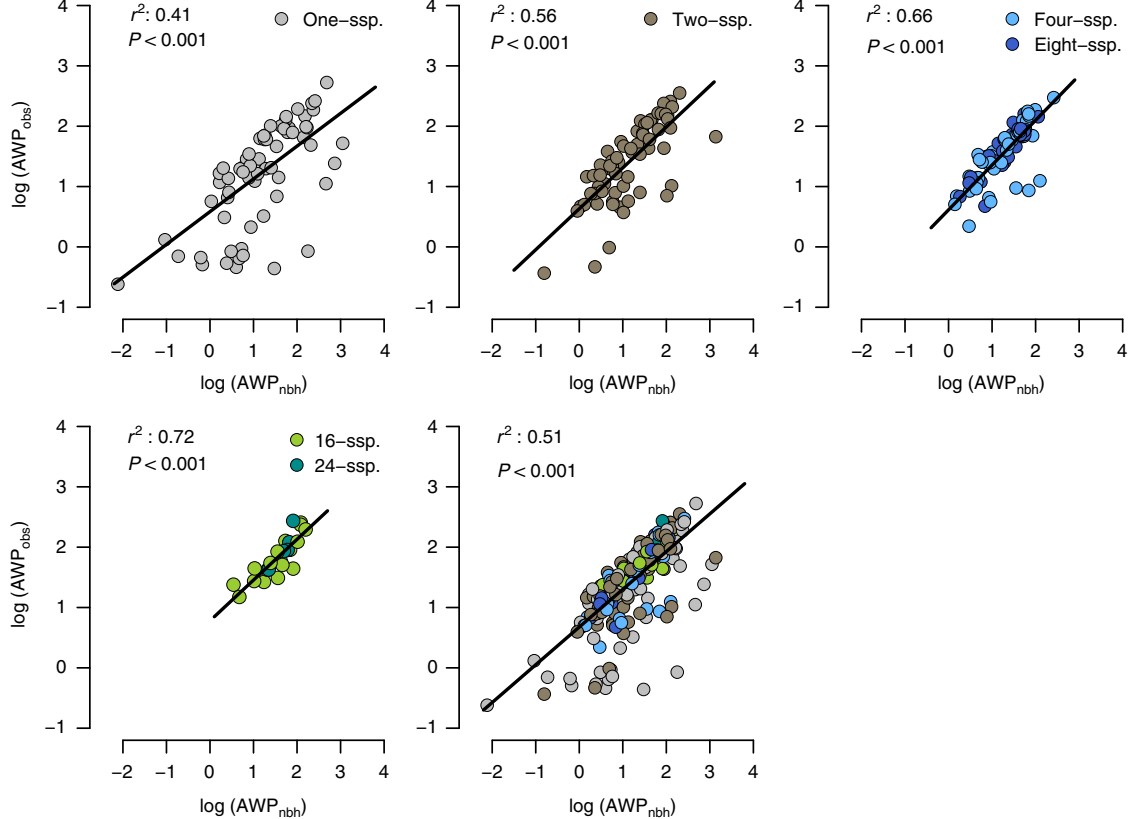

**Fig. 3** Variation in the effects of neighbourhood interactions on community productivity between species mixtures. Standardised aboveground wood productivity (AWP, $cm^3$ $cm^{-3}$ $year^{-1}$) is based on observed ($AWP_{obs}$) and predicted ($AWP_{nbh}$) annual wood volume growth of all focal trees within a plot. Note that $AWP_{nbh}$ represents the net effect of aggregated neighbourhood interactions on community productivity as predicted by a neighbourhood model. Lines represent mixed-effects model fits for monocultures, species mixtures and across all monocultures and species mixtures: low (two-species mixture), medium (four- and eight-species mixtures) and high (16- and 24-species mixtures) level of community species richness (CSR). Marginal $r^2$ values (fixed effects only) are shown for each CSR level

neighbourhood interactions ($AWP_{nbh}$) and CSR, and a negative effect of elevation (Supplementary Fig. 4). Both fixed and random effects accounted for a large proportion of the variance in $AWP_{obs}$ (fixed effects: 57% fixed and random effects: 80%; Supplementary Table 4). We found that the vast majority of the variance in $AWP_{obs}$ was explained by $AWP_{nbh}$ (52.1%), followed by community species composition (CSC; 23.2%), which was specified as a random effect in the model (see 'Methods'). In contrast, the explanatory power of log-CSR (2.0%) and heterogeneity in topography (2.5%) was extremely low (Fig. 2b). Note that the amount of variance explained by our predictors reflects partial effects, meaning the fraction attributable to each variable in the model after accounting for the effects of the other variables in the model. Interestingly, the importance of neighbourhood interactions as the predictor of community productivity was distinctly higher in species-rich (i.e., four/eight and 16-/24-species mixtures) than in species-poor communities (i.e., monocultures and two-species mixtures). This was reflected by the coefficients of determination ($r^2$), which increased consistently with CSR and ranged between 0.41 and 0.72 (values for monocultures and 16-/24-species mixtures, respectively; Fig. 3).

## Discussion

This study provides insights into processes that generate BPRs in tree communities. First, our findings provide experimental evidence that neighbourhood interactions play a fundamental role in regulating BPRs in young subtropical forests, and confirm predictions from simulation models for tropical forests[24]. Second, we found that the importance of neighbourhood interactions in regulating community productivity increased with increasing tree species richness at the community scale. Overall, these results suggest that the positive effects of biodiversity on forest productivity are primarily associated with local neighbourhood species interactions rather than processes operating at the community scale.

We found that the positive effects of species richness on community productivity were primarily driven by species interactions at the neighbourhood scale. Neighbourhood interactions might not only be related to the diversity of neighbouring trees, but also to the abundance of local competitors and focal tree characteristics (i.e., tree size and functional traits), which in turn determine its sensitivity to competition by local neighbours[35,36]. Indeed, our results demonstrate that the magnitude of positive biodiversity effects at the neighbourhood scale largely varied with initial focal tree size and NCI, where the benefits of growing in heterospecific neighbourhoods were most evident for smaller trees experiencing low competitive neighbour effects (i.e., low level of NCI; Fig. 1b). This response is most likely the result of competitive reduction due to niche differentiation among neighbours[24,35], which is particularly relevant for small individuals with a relatively low competitive tolerance[37]. Size-mediated competition tolerance is particularly evident for aboveground tree interactions, meaning that larger trees capture disproportionally greater amounts of light relative to their size when interacting with smaller ones (asymmetric competition)[38]. Given that NCI

captures the net competitive effects of neighbours larger than the focal tree in our study (asymmetric neighbourhood competition, see Methods), the main effect of an increasing NCI is most likely an increasing degree of competition for light[37,39]. However, it should be noted that larger neighbours may also have negative effects on belowground growth of a focal tree, brought about by, e.g., water and nutrient pre-emption, due to the neighbours' disproportionate advantage to access available soil resources[40]. The fact that for smaller trees, positive neighbourhood-scale biodiversity effects declined as NCI increased are therefore an indication that the relative competition intensity via (light) resource depletion becomes stronger and counteracts the positive effects of competitive reduction via, e.g., niche partitioning of canopy space[21,41], and thus, shapes the net effect of co-occurring interactions. This interpretation is supported by findings that identified competition for light as a key determinant in shaping the outcome of BPRs in forests[42,43], and that revealed stronger complementarity effects for smaller than for larger trees[43,44]. Next to competitive reduction, species may benefit from heterospecific facilitation[16,45]. For example, facilitative neighbour effects, via an improvement of microclimate conditions, were identified as a key mechanism for positive diversity effects of conservative species (e.g., species with high leaf toughness and low specific leaf area)[10]. This could explain the observed positive effect of NCI on the magnitude of neighbourhood-scale biodiversity effects as trees were larger in size (Fig. 1c), although our results reflect an across-species response. These results illustrate that both competitive reduction and facilitation—brought about by heterospecific neighbours—are fundamental mechanisms that regulate BPRs at the community scale.

The second largest proportion of variance in community productivity was explained by community species composition. Although overyielding in species mixtures was mainly driven by neighbourhood interactions, selection effects seemed to be a further important determinant of BPRs in young tree communities[5]. In contrast, the relatively low explanatory power of CSR on community BPR, after accounting for the effects of neighbourhood interactions, suggests that processes driving community BPRs, such as positive aboveground–belowground interactions[46,47] or negative density dependence of pathogens and herbivores[48,49] are particularly important at the local neighbourhood scale. Similarly, heterogeneity in topography (i.e., variation in elevation) was a weak determinant of the observed community overyielding. This is consistent with findings from site A of the experiment, where environmental variation in topography and soil chemical properties jointly only explained at maximum 4% of tree growth rates (i.e., radial crown increment)[50]. Finally, part of the unexplained variance in community BPR might be associated with litter-mediated tree interactions[51,52] or variation in leaf bacterial diversity[53], all mechanisms that have been proposed to drive overyielding, but were not considered in this study. Moreover, small-scale spatial heterogeneity in nutrient and water supply potentially affects BPRs[54]. However, given the large number of plots with varying species and species combinations in our experiment, it is less likely that the spatial configuration of plots strongly influences the outcome of BPRs. Additionally, species and species richness levels were randomly assigned to planting positions and plots[34]; thus, the likelihood that biodiversity effects were confounded with differences in belowground resource availability is relatively small.

A further important finding was that the explanatory power of neighbourhood interactions for community productivity increased with community species richness. Unsurprisingly, neighbourhood species richness tended to be higher in species-rich communities (Supplementary Fig. 1). In this case, however, it is important to note that neighbourhood species richness effects

were both size- and competition-dependent in our study (three-way interaction: $t = 2.68$, $P = 0.007$; Supplementary Table 2). Thus, the role of neighbourhood interactions in regulating overyielding at the community scale cannot be entirely attributed to the number of heterospecific neighbours. Instead, diverse neighbourhoods can modulate the mode (competition or facilitation) and intensity of local tree interactions, and thereby the strength of positive (facilitative) and negative (competitive) neighbour effects[10]. In this context, our results suggest that neighbourhood interactions become increasingly important in explaining community BPRs as CSR increases, meaning that processes leading to competitive reduction and/or facilitation at the local neighbourhood scale are fundamental in regulating the productivity of (highly) diverse tree communities.

Our results have important implications for understanding and predicting forest productivity in response to global biodiversity loss. A meta-analysis has shown positive BPRs in forests at the global scale[4]. Here, we show that tree interactions at the neighbourhood scale largely determine the growth response of tree communities to species mixing. This implies that diversity-mediated interactions among local neighbours are highly relevant for enhancing productivity in mixed-species forests— particularly in highly diverse forest communities such as subtropical or tropical ecosystems. This also highlights the importance of mixing tree species at the smallest spatial scale (i.e., the local neighbourhood level) instead of mixing monospecific patches or forest stands at the stand or landscape scale, respectively. Overall, this underlines the functional importance of local-scale species interactions in plant communities.

## Methods

**Study site and experimental design.** In this study, we used data from two spatially explicit experimental sites (site A and site B, each ~25 ha in size and ~5 km apart from each other) established in southeast subtropical China (29.08°–29.11° N, 117.90°–117.93° E) as part of the BEF-China tree diversity experiment[34]. The study sites are located on a sloped terrain (average slope 27.5° for site A and 31° for site B) between 100 and 300 m a.s.l.; the mean annual temperature is 16.7 °C and mean precipitation is 1821 mm year$^{-1}$. The predominant soil types are Cambisols, Regosols and Colluvissols[55], and the natural vegetation in the study area is characterised by subtropical mixed broad-leaved forests with a high abundance of evergreen species[56].

The experiment covers a long diversity gradient ranging from monocultures to 24-species mixtures, which were planted based on a total species pool of 40 native broad-leaved tree species (Supplementary Table 1). To ensure that all species were equally represented along the species richness gradient, species compositions of the mixtures were selected using one random (based on a 'broken-stick' design) and two non-random (based on either rarity or SLA of the species) extinction scenarios (see ref. [34]). In total, we used 474 (site A: $n = 240$, site B: $n = 234$) study plots (25.8 × 25.8 m), which were established on sites of a former *Pinus massoniana* Lambert and *Cunninghamia lanceolata* (Lamb.) Hook commercial plantation that was harvested at a rotation age of 20 years. Plots were planted in March 2009 (site A) and 2010 (site B) with 400 trees (20 × 20 individuals) using a planting scheme with equal projected distances of 1.29 m. At the time of planting, all saplings had the same age between 1 and 2 years[34]. Replanting of saplings that died during the first growing season was conducted in November 2009 (deciduous species) and March 2010 (evergreen species) at site A and 1 year later at site B. Weeding was conducted twice (2009–2011) and later once a year (since 2012) during the growing season (May–October), where all herbaceous and non-planted woody species, as well as resprouts of the previously planted *P. massoniana* and *C. lanceolata* were carefully removed[34]. Study plot species richness ranged from monocultures ($n = 150$) to mixtures of 2 ($n = 134$), 4 ($n = 91$), 8 ($n = 52$), 16 ($n = 37$) and 24 ($n = 10$) species. Species and species richness levels were randomly assigned to planting positions and plots, respectively[34].

**Tree data.** Tree measurements started in autumn 2010 (site A) and 2011 (site B) to avoid confounding effects between experimental treatments and planting. For all trees within a plot, species identity, stem diameter (measured 5 cm above the ground) and tree height (measured from the stem base to the apical meristem) were recorded in 2010 (site A) or 2011 (site B) and each subsequent year (September–October; Supplementary Table 5). To account for edge effects, growth analyses were focused on 6980 trees in the centre of the 474 study plots (hereafter: focal trees; site A: $n = 3962$, site B: $n = 3018$) that survived during the 5-year (2011–2016) study period (i.e., tree measurements were available in 2011 and

2016). The number of recorded focal trees depended on species mixture and varied between 16 (monocultures and two-species mixtures) and 100 individuals (for 4-, 8-, 16- 24-species mixtures; Supplementary Fig. 5). In 2016, a subset of 23% (site B) to 26% (site A) of the study plots were treated according to the procedure described above, while in all other plots and species mixtures, respectively, 16 central trees were used as focal trees. Trees of the outermost row of the centre within a study plot were regarded as neighbour-only trees ($n = 6793$; site A: $n = 3708$, site B: $n = 3085$; Supplementary Fig. 5). Aboveground tree–tree interactions were obvious already after 2 years of planting[57].

**Calculation of individual tree growth**. We used the annual aboveground wood volume growth ($G$; cm$^3$ year$^{-1}$) as a measure for individual tree growth. For each focal tree, we approximated the wood volume ($V$) by using a fixed value of 0.5 for form factor (i.e., a reduction factor that reduces the theoretical volume of a cylinder to tree volume[58]), which is an average value for young subtropical trees obtained from terrestrial laser scan data (Kunz et al., unpublished data; $V = (\pi D^2 / 4) * H * f$, where $D$ is the measured ground diameter, $H$ is the measured tree height and $f$ is a cylindrical form factor). $G$ was calculated from diameter and tree height measurements recorded in 2011 and 2016 (i.e., the common census interval for sites A and B)

$$G = \frac{V_2 - V_1}{t_2 - t_1} \quad (1)$$

where $V_1$ and $V_2$ are the tree wood volumes at the beginning ($t_1$) and end ($t_2$) of the study period 2011–2016. To avoid potential biases in tree-level and plot-level estimates, we excluded trees with negative growth rates in the subsequent analyses (site A: 1.7%, site B: 2.1%) that can result from, e.g., measurement errors, different measurement positions between the censuses (e.g., due to trees with trunk irregularities) or mechanical tree damage (e.g., due to falling large-sized branches)[59].

**Neighbourhood-scale model**. We used linear mixed-effects models to explore how local biodiversity patterns were modified by initial focal tree size (wood volume) and local neighbourhood conditions. The latter were characterised as the abundance of competitors (expressed as the neighbourhood competition index, NCI) and number of heterospecific (different species identity as the focal tree) tree species (NSR) in the local neighbourhood of a focal tree. The effect of NSR on individual tree growth may also depend on the number of conspecific neighbours[60]. As the number of conspecific (same species identity as the focal tree) neighbours varied within a given NSR level in our study, we used conspecific neighbour density (CND) as an additional predictor to separate the effects of CND and NSR on focal tree growth. For each focal tree $i$, NCI was calculated as the total basal area of closest neighbours $j$ with a larger stem diameter than the focal tree ($\sum_{j \neq i} \pi D_j^2 / 4$, where $D$ is the measured ground diameter), CND as the total number of closest conspecific neighbours and NSR as the total number of closest heterospecific neighbour species ($\sum_{j \neq i} N_j$, where $N$ is the recorded species number). Both NCI, CND and NSR represent the net effect of neighbouring trees on the growth of a focal tree. Although neighbour effects can be size-symmetric (i.e., summed basal area of all neighbours) or size-asymmetric (i.e., summed basal area of neighbours with a larger stem diameter than the focal tree)[61], preliminary analysis indicated that NCI based on asymmetric competition provided a significant better fit to the data compared to the size-symmetric NCI ($\Delta$AIC $= 426.5$, $P < 0.001$). Given the close correlation between neighbour tree diameter ($D$) and height ($H$) in this study (Pearson correlation: $r = 0.91$, $P < 0.001$; Supplementary Fig. 6), larger neighbours were assumed to be taller. On this basis, we examined the changes in annual wood volume growth of a focal tree as a basic function of its size and local interactions with neighbouring trees based on NCI, CND and NSR. Tree size and NCI were log$_{10}$-transformed to linearise their relationship with annual growth rate (see ref. 36 for a related approach). The basic model had the form

$$\log(G_{i,j,s,k,p}) = \alpha + \beta_1 \log(V_{i,j,s,k,p}) + \beta_2 \log(\text{NCI}_{i,j,s,k,p} + 1) + \beta_3 \text{ CND}_{i,j,s,k,p} + \beta_4 \text{ NSR}_{i,j,s,k,p} + \gamma_j + \varphi_s + \upsilon_k + \tau_p + \varepsilon_{i,j,s,k,p} \quad (2)$$

where $G$ is the annual wood volume growth over a 5-year interval of focal tree $i$ of species $j$ growing in neighbourhood condition $s$ (species composition) and $k$ (total number of neighbours) in plot $p$; $\alpha$ is the intercept and $\beta_{1,2,3,4}$ are parameters adjusting the effects of initial focal tree wood volume ($V$), neighbourhood competition (NCI), conspecific neighbour density (CND) and neighbourhood tree species richness (NSR); $\gamma$, $\varphi$, $\upsilon$ and $\tau$ denote crossed random effects of focal trees' species identity, neighbourhood species composition, neighbour density and plot identity, respectively, and $\varepsilon$ is the residual error—assuming a normal distribution with mean 0 and variance $\sigma^2$ of all variance components. We included plot, species identity and neighbourhood species composition in the random structure to account for variation in abiotic growing conditions within a study site (e.g., small-scale differences in topography) and species-specific effects. We also tested for a random effect that allowed the effects of NSR to vary among species, but found no statistical support for such a random slope model, demonstrating that the shape of the $G$-NSR relationship was consistent across species ($\chi^2 = 2.00$, $P = 0.367$). Due to mortality of re-planted trees, we used the average values of NCI, CND and NSR in

the study period (2011–2016), as they most accurately reflect the neighbourhood conditions experienced by a focal tree during the observation period[37]. The average mortality rate across study species in the study period (focal and neighbour-only trees) ranged between 17% (site A) and 23% (site B).

First, we determined the optimal random-effects structure based on restricted maximum likelihood (REML) estimation, including all covariates and possible interaction terms. Second, we determined the optimal fixed-effects structure by using the maximum likelihood (ML) method[62]. Different competing models (including all possible combinations of covariates and two- and three-way interaction terms with NSR) were evaluated by sequential comparison based on the Akaike information criterion (AIC). The model with the lowest AIC and highest Akaike weights (i.e., the likelihood of being the best-fitting model based on AIC values)[63], respectively, was chosen as the most parsimonious model (Supplementary Table 6). We further simplified the model with the lowest AIC by removing all terms that were not significant according to likelihood ratio tests[62]. Parameter estimates of the best-fitting model were based on restricted maximum likelihood (REML) estimation[62] and are presented in Supplementary Table 2. All predictors were standardised (divided by their standard deviations) before analysis. Models were fitted for each study site separately. There was no critical correlation between covariates (collinearity), as indicated by the variance inflation factors (all VIFs <2.7). Model assumptions (including spatial independence) were checked and confirmed according to ref. 62.

**Quantifying neighbourhood-scale net biodiversity effects**. To examine how local biodiversity effects were related to neighbourhood species richness, we used growth predictions (based on fixed-effects estimates) from our best-fitting model (Supplementary Table 2). The individual-based net biodiversity effect at the neighbourhood scale (NE$_{nbh}$) for a given NSR-level $j$ was calculated as the relative change in annual wood volume growth ($G$) of a focal tree growing in conspecific (NSR $= 0$) compared to heterospecific neighbourhoods (NSR $= 1,...,7$)

$$\text{NE}_{nbh,j} = 100 \frac{G_{h,j} - G_c}{G_c} \quad (3)$$

where c indicates conspecific and h heterospecific neighbours with $j = 1,...,7$ species. NE$_{nbh,j}$ was then related to species richness of the local neighbourhood, separately for low, average and high NCI. For each focal tree, we predicted $G$ at low (20% quantile of log-transformed NCI), average (50% quantile) and high (80% quantile) abundance of competitors in its local neighbourhood. We did this for every level of NSR, while keeping the tree size fixed at a specific value using the 20% (small trees), 50% (medium-sized trees) and 80% (large-sized trees) quantile of log-transformed initial wood volume. In this way, our function-derived growth rates allowed us to analyse how neighbourhood-scale biodiversity effects vary with NSR.

**Sensitivity analysis**. Given the negative correlation between NSR and CND in our study ($r = -0.76$), neighbourhoods with a high number of heterospecific species are associated with fewer conspecific neighbours. We therefore compared the relative importance of NSR and CND effects by fitting a series of candidate models for each predictor separately. We found strong statistical support that NSR is an important driver regulating individual tree productivity rather than CND, because CND was not significant (Supplementary Table 7).

To assess whether our results depend on the calculation of the neighbourhood competition index (NCI), we ran a series of candidate models either using size-symmetric (i.e., summed basal area of all neighbours) or size-asymmetric (i.e., summed basal area of neighbours with a larger stem diameter than the focal tree) NCIs. We found qualitatively similar results (Supplementary Tables 6 and 8), but the inclusion of size-asymmetric NCI effects into the best-fitting model resulted in a substantial drop of AIC (size-asymmetric NCI: 6352.7; size-symmetric NCI: 6536.8), and the Akaike weights indicated that the model including size-asymmetric NCI effects has a relative likelihood being the best-fitting model of 100% compared to the model including size-symmetric NCI effects.

**Calculation of community productivity**. In this study, communities are defined as the total number of focal trees within a given plot. For each plot of site B, the aboveground wood productivity (AWP) was calculated based on individual tree growth (annual wood volume growth; $G$) of all focal trees within a plot. The contribution of a given tree to AWP strongly depends on its initial size[64]. Thus, differences in size structure among species mixtures might cause spurious correlations between community productivity and species richness when individual tree growth rates are scaled up to plot-level productivity. We therefore considered the relative importance of each focal tree in terms of its contribution to the total mean wood volume (see ref. 65 for a related approach). We used the total mean wood volume (2011–2016) instead of the total initial wood volume (2011) to account for potential bias associated with differences in tree density (i.e., the number of trees per plot that can vary with the sampling scheme or mortality; see sections above). Observed community productivity (AWP$_{obs}$) was quantified as

$$\text{AWP}_{obs} = \frac{\sum_{i=1}^N (G_{obs,i} * V_i)}{\sum_{i=1}^N \overline{V}_i} \quad (4)$$

where $AWP_{obs}$ is the observed annual standardised aboveground wood productivity ($cm^3\ cm^{-3}\ year^{-1}$) of a given plot, and $G_{obs,i}$, $V_i$ and $\overline{V}_i$ are the observed annual wood volume growth, initial wood volume (2011) and mean wood volume in the study period 2011–2016 of focal tree $i$, respectively. Similarly, we calculated community productivity based on predictions from our neighbourhood model ($AWP_{nbh}$). Here, we used parameter estimates obtained from our best-fitting neighbourhood model for trees growing at site A (Supplementary Table 2) to predict the annual wood volume growth ($G$) of all focal trees growing at site B, meaning that we related parameter estimates—derived from site A—directly to observed focal tree and neighbour data of site B

$$AWP_{nbh} = \frac{\sum_{i=1}^{N}(G_{nbh,i} * V_i)}{\sum_{i=1}^{N}\overline{V}_i} \tag{5}$$

where $AWP_{nbh}$ is the predicted standardised annual aboveground wood productivity ($cm^3\ cm^{-3}\ year^{-1}$) of a given plot based on tree interactions at the neighbourhood scale. $G_{nbh,i}$ is the predicted annual wood volume growth of focal tree $i$ using parameter estimates of a neighbourhood model (site A), and $V_i$ and $\overline{V}_i$ are the observed initial wood volume (2011) and mean wood volume in the study period 2011–2016 of focal tree $i$, respectively. Note that mean mortality rates across species did not substantially differ among species richness levels (Supplementary Table 9).

**Community-scale model**. We used linear mixed-effects models to determine drivers of the biodiversity–productivity relationship (BPR) at the community scale. To account for variation in tree species composition among study plots, plot species composition was used as a random effect. $AWP_{obs}$ was used as a response variable and fixed effects were included for community tree species richness (CSR), community productivity based on neighbourhood inteactions ($AWP_{nbh}$) and for small-scale variation in topography (elevation, slope and 'northness') among study plots within a study site. For each plot, data on mean elevation (m), slope (°) and 'northness' (cosine-transformed radian values of the aspect) were extracted from a 5-m digital elevation model (DEM) based on differential GPS measurements. The overall quality of the DEM was high, with an explained variance of 98% and a root mean square error (RMSE) of 1.9 m (10-fold cross-validation) in an elevation range of 112 m (see ref. [57]). Model selection was based on the procedure as described above for the neighbourhood models. The response variable, CSR and $AWP_{nbh}$ were $log_{10}$-transformed to meet model assumptions. All predictors were standardised (divided by their standard deviations) before analysis. There was no indication for collinearity (all VIFs < 1.2). Model assumptions were checked and confirmed according to ref. [62].

To quantify the contribution of fixed- and random-effects variables in explaining variation in community productivity along the species richness gradient, we conducted a variance-partitioning analysis using the method of ref. [66] that computes the fraction of variation attributable to each variable in a regression model. Variance partitioning was performed with the best-fitting model. This analysis allowed us to quantify the importance of neighbourhood interactions in driving BPRs in young tree communities.

All analyses were conducted in R (version 3.3.1)[67] using the packages lme4[68], lmerTest[69], MuMIn[70] and variancePartition[66].

**Data availability**. Data that support the findings of this study have been deposited in the BEF-China project database (http://china.befdata.biow.uni-leipzig.de/) and are available from the corresponding authors on reasonable request.

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

## Acknowledgements

We are grateful to the many workers and students who helped to conduct the tree inventories and to all members of the BEF-China consortium who coordinated and helped with the establishment and maintenance of the experiment. We acknowledge support by the German Research Foundation and the Open Access Publication Funds of the SLUB/TU Dresden. This research was funded by the German Research Foundation (DFG FOR 891/1-3, HA 5450/1-2, BR 1698/9-3 and OH 198/2-3).

## Author contributions

G.v.O., W.H. and H.B. designed the research. Y.L. and M.K. collected and compiled the data. A.F. analysed the data and wrote the manuscript. All authors contributed to revisions.

## Additional information

**Competing interests:** The authors declare no competing interests.

