## [Peer Review File(PDF 520 kb) · Nature Communications]

Reviewers' comments:

Reviewer #1 (Remarks to the Author):

This is an interesting and potentially important paper addressing a classical question in community ecology, namely whether mixed species assemblages over-yield relative to species-poor assemblages. The analysis makes use of a large-scale experiment underway in China to study the effects of diversity on community level functioning. The authors conclude that there is indeed a species richness affect on productivity, but that it is primarily a local stand phenomenon and not due to communitywide processes. They then argue that more attention be paid to small spatial scales of interacting neighborhoods to understand the effects of species richness on community level processes. They analyze wood yield in local stands varying in species richness from one to 24 tree species. Their statistical design evaluates relative wood yield in focal trees in species rich local stands compared to yields of focal trees in monospecific stands.

This kind of analysis can be tricky, however, as our own work on Barro Colorado Island has shown us. We analyzed a similar question in a non-experimental 50 ha plot, looking at how species richness in the local neighborhood of a tree affected demographic performance of focal individuals, analyzing both survival and growth in diameter at breast height (dbh). We initially found a species richness affect in both survival and growth rate. However, the species richness affect largely disappeared when we took out the confounding effect of number of neighbors that were conspecific with the focal tree. If I understand the analysis done by the authors, I believe that the number of conspecific trees in a focal tree's neighborhood is confounded with species richness. That is, there are fewer conspecific trees in neighborhoods containing more different tree species. In all of our studies on Barro Colorado Island, we have found that effects of conspecific neighbors are vastly more important on the performance of individual trees than are interactions with interspecific neighbors.

So the authors have not yet convinced me that conspecific effects can be ruled out as the primary driver of the apparent effect of species richness. To satisfy me on this point, I ask that they do a stepwise regression in which the first variable entered is the number of conspecific trees in the neighborhood, and only after this variable is in the model, add a variable for species richness. My prediction is (and I am happy to be proven wrong) that once the number of conspecific neighbors is taken into account, the neighborhood species richness variable will not be significant or much less so. Is it not true, in this experimental design, that the number of conspecific neighbors must inevitably decline with neighborhood species richness? That is, don't the number of conspecific neighbors and species richness negatively co-vary? I can conceive of an alternative design in which only the species richness of heterospecifics is manipulated in neighborhoods of focal trees, but based on the figure in the supplemental material, this does not appear to be what was done in the case of this experiment. Therefore, it appears that the experimental design confounds the number of neighboring conspecifics with species richness. If the authors find that the number of conspecifics explains most of their apparent effect, then they will have to alter their conclusions regarding the effects of species richness on over yielding. In either case, the result will be interesting and important.

There are several other questions that I have about the methods that I could not find answered either in the paper or in the supplemental material. One was the possibility that focal trees were reused in the neighborhood analyses of other focal trees. If this is the case, then there is an issue of independence and spatial autocorrelation that is not addressed in the current analysis. Ideally, every focal tree would have been used only once with a unique neighborhood of trees not used in other neighborhoods. However, I recognize that this would require an impossibly large experimental design. So because I assume that such non-independence and spatial autocorrelation exists in the study, the statistical design must incorporate this non-independence and spatial autocorrelation to be valid.

Doing so of course reduces the degrees of freedom and therefore the confidence intervals on yield effects are broader. However, the good news is that typically including spatial autocorrelation does not change the effect size.

I have a number of more detailed questions about the methods that I would like addressed at least in the supplemental material. (1) How old and large were the trees at the point of yield measurement in the experiment? (2) How far apart were the trees planted, and were their crowns touching or interacting spatially (this affects one's conclusions about the mode of interactions going on among neighbors)? (3) Over what time interval were the yield measurements made (how many years)? (4) Did mortality reduce the number of neighboring trees in any of the neighborhoods? If so, how was mortality factored into the analysis, if at all? (5) There was huge variation in yield among the conspecific neighborhoods. Were there any traits of the different tree species that could explain this variability? For example, were the species with the poorest performance those with lower wood density and more light demanding (these species are more affected by conspecific negative density effects on BCI)? (6) The variance in AWP (Fig.1) is much greater in the one species neighborhoods than in the 24 species neighborhoods, and there is a steady drop in the variance from conspecific local stands to the most species rich local stands. What explains this pattern? Without tables of the actual species that were planted in each neighborhood, it is impossible to evaluate whether the particular mixtures of species are responsible for the drop in variance, or, as I suspect, it is because no one species has a very strong negative influence on itself because no single species has lots of conspecific neighbors in the species-rich neighborhoods, and so all focal trees in those neighborhoods perform relatively better. This would explain the systematic drop in variance in yields with increasing numbers of species. (7) I would like to know if those species that performed the poorest in conspecific stands also were the poorest performing species in the species rich stands, especially if in the latter stands they had more conspecific neighbors. (8) For a given species richness, did the number of neighbors that were conspecific with the focal tree vary at all? If so, did focal trees, for a given neighborhood species richness, perform worse if there were more than if there were fewer conspecific neighbors?

I recommend three of our papers on Barro Colorado Island results for the authors to read, listed below. In these papers we did not analyze yield but were focused on survival and dbh growth. There are other growth results scattered among a number of other papers, which I can track down if the authors care to send me an email.

Hubbell, S. P., J. A. Ahumada, R. Condit, and R. B. Foster. 2001. Local neighborhood effects on long-term survival of individual trees in a Neotropical forest. *Ecological Research* 16: S45-S61.

Ahumada, J. A., S. P. Hubbell, R. Condit, and R. B. Foster. 2004. Long-term tree survival in a neotropical forest: The influence of local biotic neighborhood. Pp. 408-432, Chapter 23 in: E. Losos, E. G. Leigh, Jr., eds. *Forest Diversity and Dynamism: Findings from a Network of Large-Scale Tropical Forest Plots*. University of Chicago Press, Chicago, IL

Uriarte, M., R. Condit, C. D. Canham, and S. P. Hubbell. 2004. A spatially-explicit model of sapling growth in a tropical forest: Does the identity of neighbours matter? *Journal of Ecology* 92: 348-360.

Review signed by Steve Hubbell

Reviewer #2 (Remarks to the Author):

In their well written manuscript entitled "Neighbourhood interactions drive overyielding in mixed-species tree communities", Fichtner and colleagues describe how local neighbourhood interactions

between trees drive positive relationships between plot scale species richness and aboveground productivity.

The finding that neighbouring trees can promote the growth of a focal tree is not new, and has also been described in various other studies (although in the introduction section, the authors do suggest that there is a lack of published research on this). What distinguishes this work from other studies, is that as a second step, the authors investigated how these local diversity effects on individual tree growth scale up to the level of communities, and hence help explaining community level diversity - productivity relationships as observed in many forests. They conclude that "diversity-mediated interactions among local neighbours are more relevant for enhancing productivity in mixed-species forests than community tree species richness per se". While this is of course a very interesting conclusion, I am not yet convinced that it is supported by the analyses, as this conclusion requires the testing of more 'alternative hypotheses'.

Individual tree growth is statistically modelled as a response of the identity and richness of up to eight 'neighbouring' trees. However, what would happen if instead, individual tree growth is statistically modelled as a response of the identity and richness of more (within a larger radius), or fewer (e.g. only the closest) neighbouring trees? The authors could run various alternative models, differing in the spatial scale at which 'neighbourhood' is defined. (as a side note, I would like to note that it was not entirely clear to me how 'neighbourhood' was actually defined: as the tree individuals that are within a certain (e.g. 5m) radius of the focal tree? How was it decided whether 1, 2, 3 ... or 8 individuals were included?) The authors could then specifically look which type of neighbourhood is most relevant in driving tree growth. Only then, conclusions about the scale at which diversity matters most can be drawn.

I also have the comment regarding the presentation of results. The analyses consisted of two main parts: (1) the analysis of focal tree growth and (2) the analysis of community level tree productivity. While the results of the second part are clearly presented in the figures, the results of the first part are only described in the text and in the supplementary material, but not illustrated with figures in the main document. To really convince readers of the relevance of the findings, I think it would help to add at least one figure that illustrates the results of the first part of the study. For example, a graph showing the relationship between over yielding of focal tree growth and neighbourhood species richness (such as supplementary figure 1, but preferably also including the data points or at least the R^2 , which would help assessing the strength of this relationship) would be extremely helpful. Regarding Fig. S3: I found it slightly concerning the standardised effect of neighbourhood AWP is above one. While this is theoretically possible, standardised effects with absolute values greater than one are often a sign of multicollinearity. I realise that the authors tested for a multicollinearity and reported low VIF values, but I do wonder whether this extremely high standardised effects might have been caused by a strong relationship between neighbourhood AWP and one of the random effects? I urge the authors to have a serious look at this, as such extremely standardised effects may be a sign of a flawed model.

Additional comments

L 30-31: meta-analyses such as Cardinale et al, 2011, *Am. J. Bot.* suggest something else, namely that biodiversity effects are largest at large scales, and suggest that this may be due to large scale spatial heterogeneity in e.g. resources

L 43-48: doesn't the sentence of line 38-40 suggest that neighbourhood and community richness should both be preserved, as they strengthen each other's effects? In that case, the conclusion needs to be rephrased.

L 54-56: there is actually quite a long list of studies that investigated (taxonomic / functional / phylogenetic) diversity-productivity/growth relationships at the neighbourhood scale: e.g. Potvin &

Gotelli, 2008, Uriarte et al, 2010; Lubbe et al, 2015, Front. In Plant Sci.; Kunstler et al, 2016, Nature; Ecol. Lett.; del Rio et al, 2017, J. of Ecol; Dillen et al, 2017, Forest Ecol. & Managm. and the authors own, previous work (Fichtner et al, 2017, Ecol. Lett), to name just some articles (I am sure there are more).

L 61: niche differentiation is a bit vague, as niches can be very broadly defined. So I would suggest 'resource partitioning' as an alternative

L 65-66: it should be added that these well performing species also need to dominate in order to get positive selection effects.

L 94-98: rephrase the sentence, as there is a lot of redundancy within it. Currently, it is basically stated twice what are your objectives

L 99: so you basically expect a positive interaction effect between neighbourhood and plot-scale richness? Why?

L115: add 'and compare' after 'obtain'

L 137: what is meant with 'consistently'?

Fig. 2: the panels of this figure are too small, making it almost impossible to read. I suggest to rearrange it into a 2x2 panel figure.

L 187: local -> neighbourhood

L 203-205: quite likely, aboveground size is highly related to belowground size. In that case, results could be just as well be driven by belowground competition for e.g. nutrients or water, especially in deeper soil layers.

L 211: add 'a' after 'as'

L 214-217: please rephrase, this sentence is hard to understand. Also, given the overall negative NCI effects, is it really likely that facilitation plays a key role in this study? Or is competition simply weakened by species richness?

L 239-243: unclear sentence, please rephrase

L 276: site -> sites

L 313: aren't their more than 2 measurements per tree, as measurements were done each year from 2011-2016?

L 325: what is 'form factor'?

L 333-337: as a sensitivity analysis, I would like to see how this choice affected the results

L 344-347: as a sensitivity analysis, it would be good to also quantify NCI based on the biomass of all individuals, including those that are smaller than the focal tree, as it is possible that smaller individuals also have competitive effects

L 356-359: it would be good to actually test whether these relationships are indeed linear, which could be done by also running an alternative model with additionally a quadratic term, and to investigate whether the simpler, linear model is more parsimonous than the model also including a quadratic term

L 389-390: doesn't this contradict L383-384? Please clarify

L 391: sites -> site; add "strong" before "correlation"

L 438: given that all plots were very close to each other, how accurate are the GPS measurements in describing local scale variation in altitude, northness and slope?

Supplementary material

Fig. S1, left panel: how can the diversity effect be above 0 when species richness is 1? Also, the over yielding effects seem relatively small (up to ~10%). How come that the community level diversity effects are much larger?

We are grateful to the reviewers for their encouraging and constructive comments to improve the manuscript. We have considered all the suggestions made by the reviewers and have revised the manuscript accordingly. We also revised the reference section by including new references suggested by the reviewers and recent publications.

Please find below a detailed point-by-point response and a description of the changes (each referring to page and line numbers in the revised manuscript “R1_clean edit”) related to the comments of the two reviewers.

Reviewers' comments:

Reviewer #1 (Remarks to the Author): This is an interesting and potentially important paper addressing a classical question in community ecology, namely whether mixed species assemblages over-yield relative to species-poor assemblages. The analysis makes use of a large-scale experiment underway in China to study the effects of diversity on community level functioning. The authors conclude that there is indeed a species richness affect on productivity, but that it is primarily a local stand phenomenon and not due to communitywide processes. They then argue that more attention be paid to small spatial scales of interacting neighborhoods to understand the effects of species richness on community level processes. They analyze wood yield in local stands varying in species richness from one to 24 tree species. Their statistical design evaluates relative wood yield in focal trees in species riche local stands compared to yields of focal trees in monospecific stands.

This kind of analysis can be tricky, however, as our own work on Barro Colorado Island has shown us. We analyzed a similar question in a non-experimental 50 ha plot, looking at how species richness in the local neighborhood of a tree affected demographic performance of focal individuals, analyzing both survival and growth in diameter at breast height (dbh). We initially found a species richness affect in both survival and growth rate. However, the species richness affect largely disappeared when we took out the confounding effect of number of neighbors that were conspecific with the focal tree. If I understand the analysis done by the authors, I believe that the number of conspecific trees in a focal tree's neighborhood is confounded with species richness. That is, there are fewer conspecific trees in neighborhoods containing more different tree species. In all of our studies on Barro Colorado Island, we have found that effects of conspecific neighbors are vastly more important on the performance of individual trees than are interactions with interspecific neighbors. So the authors have not yet

convinced me that conspecific effects can be ruled out as the primary driver of the apparent effect of species richness. To satisfy me on this point, I ask that they do a stepwise regression in which the first variable entered is the number of conspecific trees in the neighborhood, and only after this variable is in the model, add a variable for species richness. My prediction is (and I am happy to be proven wrong) that once the number of conspecific neighbors is taken into account, the neighborhood species richness variable will not be significant or much less so. Is it not true, in this experimental design that the number of conspecific neighbors must inevitably decline with neighborhood species richness? That is, don't the number of conspecific neighbors and species richness negatively co-vary? I can conceive of an alternative design in which only the species richness of heterospecifics is manipulated in neighborhoods of focal trees, but based on the figure in the supplemental material, this does not appear to be what was done in the case of this experiment. Therefore, it appears that the experimental design confounds the number of neighboring conspecifics with species richness. If the authors find that the number of conspecifics explains most of their apparent effect, then they will have to alter their conclusions regarding the effects of species richness on over yielding. In either case, the result will be interesting and important.

Response: We are grateful for the point made by the reviewer and followed his suggestions. We agree that conspecific neighbour density could be a strong growth determinant and included this information in the main text (p15, L323-331). As suggested, we also included conspecific neighbour density (CND) as an additional predictor in our model and adjusted all our results (e.g. equation 2). To carefully account for the reviewer's recommendation, we tested for CND effects via three different approaches, but we found no statistical support for a CND effect on individual tree growth.

1. We were primarily interested in determining systematically, which predictors (initial tree size V , neighbourhood competition index NCI , conspecific neighbour density CND , neighbourhood species richness NSR) and which predictor combination would result in a most parsimonious model, without having a priori expectations on the importance of predictors (i.e. sequential order of predictors) for individual tree growth. Therefore, we used a set of 25 candidate models containing all possible predictor combinations including two-way and three-way interactions of NSR with V , NCI and CND in the candidate models (see Supplementary Table 6 in the revised version). All candidate models were compared according to the Akaike Information Criterion (AIC) – the most common approach for model selection in recent ecology research

(Burnham & Anderson 2002). We selected the model with the lowest AIC as the most parsimonious model and further simplified the model with the lowest AIC by removing all terms that were not significant according to likelihood ratio tests. This resulted in a best-fitting model including effects of V, NCI and NSR and their 3-way interaction. We found, thus, no statistical support for a CND effect on individual tree growth (χ^2 : 0.37, $P = 0.540$; p7 L128 and Supplementary Table 6). All other candidate models resulted in an AIC difference larger than 94 and Akaike weights (indicating the relative likelihood of each model being the best-fitting model) of 0%, thus indicating no empirical support (Δ AIC values greater than 20 have essentially no empirical support according to Burnham *et al.* 2011; see Supplementary Table 6).

2. Although the variance inflation factors (VIFs) indicated no serious collinearity between predictors (all VIFs < 2.7; Zuur *et al.* 2009), NSR and CND were negatively correlated in our study ($r = -0.76$). To test the robustness of our results, we therefore ran a sensitivity analysis by fitting a series of candidate models for each predictor separately. We found strong statistical support that NSR is a more important driver regulating individual tree growth rather than CND, because in contrast to NSR, CND effects were not significant (χ^2 : 1.21, $P = 0.271$; see new Supplementary Table 8). We included this information in a new section “Sensitivity analysis” in the Supplementary Methods.
3. To further convince the reviewer, we also applied a sequential testing approach (Type I ANOVA). As suggested, we first entered CND before adding NSR. Similarly, the effect of CND was not significant (Table R1).

Table R1: Effects of tree size (V), neighbourhood competition (NCI), conspecific neighbour density (CND) and neighbourhood species richness (NSR) on individual tree growth. *F*-values refer to Type I ANOVA with Satterthwaite approximation for degrees of freedom.

Fixed effect	d.f.	F -value	P -value	Fixed effect	d.f.	F -value	P -value
Model 1							
V (log)	3878.2	2755.5	< 0.001	V (log)	3878.2	2755.5	< 0.001
NCI (log+1)	1958.1	118.7	< 0.001	CND	500.4	2.6	0.108
CND	492.3	1.1	0.305	NCI (log+1)	2183.7	112.7	< 0.001
NSR	708.2	4.0	0.045	NSR	708.2	4.0	0.045

There are several other questions that I have about the methods that I could not find answered either in the paper or in the supplemental material. One was the possibility that focal trees were reused in the neighborhood analyses of other focal trees. If this is the case, then there is an issue of independence and spatial autocorrelation that is not addressed in the current analysis. Ideally, every focal tree would have been used only once with a unique neighborhood of trees not used in other neighborhoods. However, I recognize that this would require an impossibly large experimental design. So because I assume that such non-independence and spatial autocorrelation exists in the study, the statistical design must incorporate this non-independence and spatial autocorrelation to be valid. Doing so of course reduces the degrees of freedom and therefore the confidence intervals on yield effects are broader. However, the good news is that typically including spatial autocorrelation does not change the effect size.

Response: Yes, focal trees were re-used as neighbours for other focal trees. We therefore tested for spatial autocorrelation, following Zuur *et al.* (2009), by analysing whether the residuals of our best-fitting neighbourhood model (Supplementary Table 2) show any spatial dependence. However, we found no evidence that the independence assumption was violated (Fig. R1). Accordingly, we revised the ‘model assumption sentence’ by including that we tested for spatial autocorrelation (p17 L381).

Fig. R1: Semi-variogram of the standardised residuals obtained by our best-fitting mixed-effects model. Note that there is no spatial correlation up to a certain distance. As “*spatial dependence shows itself as an increasing*

band of points, which then levels off at a certain distance” (Zuur et al. 2009), there is no indication for spatial correlation in the data.

I have a number of more detailed questions about the methods that I would like addressed at least in the supplemental material.

(1) How old and large were the trees at the point of yield measurement in the experiment?

Response: At the time of planting, all seedlings had the same age between 1 and 2 years, resulting in an average age of 8-9 years in 2016 (end of the study period). Average height in 2016 varied between 3.6 m (site B) and 4.2 m (site A). We included additional information on seedling age in the main text (p12 L267) and information on initial and final tree size (height and diameter) in the Supplementary Table 5.

(2) How far apart were the trees planted, and were their crowns touching or interacting spatially (this affects one’s conclusions about the mode of interactions going on among neighbors)?

Response: All plots were planted with 400 trees (20 x 20 individuals) at equal projected distances of 1.29 m. We rephrased this sentence to provide more information on the planting scheme (p12 L268-270). We also included additional information on initial and final tree size (height and diameter) and annual height and diameter growth rates in the Supplementary Table 5. On average, tree height was 1.30 m (maximum 5.82 m) and 4.19 m (maximum 14.80 m) at the beginning and end of the study period, respectively. Average height increment was 0.59 m per year. Thus, most of the study species were fast-growing. Given a planting distance of 1.29 m, an average initial height of 1.30 m and an average growth rate of 0.59 m/year, crown touching or crown interactions were obvious for most of the trees at the latest two years after the start of the study period. Moreover, in a previous study, using the same study site (site A), Li *et al.* (2014) found that some species already two years after planting trees were interacting aboveground, as annual growth rate (2010-2011) in crown width was significantly affected by neighbourhood competition. We added this information in the main text (p13 L296-297).

(3) Over what time interval were the yield measurements made (how many years)?

Response: We used a five-year interval (2011-2016) and rephrased several sentences throughout the manuscript to more explicitly mention the underlying study period (e.g. p13 L288, p14 L306, L311-312, p18 L402, L416).

(4) Did mortality reduce the number of neighboring trees in any of the neighborhoods? If so, how was mortality factored into the analysis, if at all?

Response: Yes, mortality reduced the number of neighbouring trees (Supplementary Table 7). We therefore accounted for neighbour mortality by using the mean values of NCI and NSR between 2011 and 2016 (p16 L362-364) following the approach of Coomes and Allen (2007).

Moreover, in the revised version of the manuscript we additionally considered a random effect in the neighbourhood model that accounts for the initial neighbour density. This random effect significantly improved the model ($P = 0.021$). Thus, we accounted for both, mortality at the beginning (i.e. neighbouring trees that died between 2009/10 and 2011) of the study period by using a random effect ‘neighbour density’ and over the course of the study period (2011-2016) by using mean values of neighbourhood competition indices, neighbourhood species richness and conspecific neighbour density. We rephrased the respective text passages accordingly.

(5) There was huge variation in yield among the conspecific neighborhoods. Were there any traits of the different tree species that could explain this variability? For example, were the species with the poorest performance those with lower wood density and more light demanding (these species are more affected by conspecific negative density effects on BCI)?

Response: As suggested, we analysed if community productivity of monocultures is related to functional traits of the component species. We selected four key traits (specific leaf area, SLA; leaf nitrogen concentration, LNC; leaf toughness, LT; wood density, WD) which are linked to productivity and shade tolerance, and which thus reflect tree ecological strategies (e.g. Valladares & Niinemets 2008; Lasky *et al.* 2014, Kröber *et al.* 2015). As expected, community productivity of monocultures was linked to functional traits of the component species such as wood density ($P < 0.01$) and leaf toughness ($P < 0.001$; see new Supplementary Figure 2). We included this information in the main text (p7 L137-139). It is worth pointing out that SLA and leaf N content had no significant effect on aboveground wood productivity, which confirms findings of previous papers that analysed growth across a large set of tree species (Paine *et al.* 2015).

(6) The variance in AWP (Fig.1) is much greater in the one species neighborhoods than in the 24 species neighborhoods, and there is a steady drop in the variance from conspecific local stands to the most species rich local stands. What explains this pattern? Without tables of the actual species that were planted in each neighborhood, it is impossible to evaluate whether the particular mixtures of species are responsible for the drop in variance, or, as I suspect, it is because no one species has a very strong negative influence on itself because no single species has lots of conspecific neighbors in the species-rich neighborhoods, and so all focal trees in those neighborhoods perform relatively better. This would explain the systematic drop in variance in yields with increasing numbers of species.

Response: The pattern higher variation in the monoculture is mainly explained by the highly different species-specific growth rates in the experiment (see Fig. S1 in Li *et al.* 2017).

Figure S1 taken from Li *et al.* (2017). Mean growth rate of wood volume for 37 tree species.

In this context it is important to mention that Figure 1a (now Fig. 2a) displays the observed growth rate at the community level based on individual tree growth of all living focal trees within a plot of site B and not local neighbourhood results. Due to the large number of different neighbourhood species compositions in a given stand (plot level), we cannot provide a table for specific neighbourhood species compositions for each of the 234 plots. However, in our neighbourhood model, we accounted for differences in species composition in the neighbourhood of a focal tree by using a random effect for neighbourhood species composition.

Given the negative correlation between NSR and CND in our study (see above), we agree that a reduction in intraspecific competition with increasing community species richness would be the main cause for the observed decline in variance in community productivity. This is actually one of the conclusions in Fichtner *et al.* (2017). Moreover, the decrease in variance of community productivity might also be related to increasing community species richness due to a statistical reason, because for a given species richness level extreme values are more extreme the larger the sample size is (Schmid *et al.* 2008).

(7) I would like to know if those species that performed the poorest in conspecific stands also were the poorest performing species in the species rich stands, especially if in the latter stands they had more conspecific neighbors.

Response: We agree with this statement of the reviewer; those species with lowest productivity in monocultures were also associated with lowest growth rates in species-rich (16- and 24- species) mixtures, but derived most benefit from growing in species-rich mixtures. We added this information in the main text (p7 L139-142) and Supplementary information (see new Supplementary Figure 3). Please note that we could not find any statistical support for an effect of conspecific neighbour density on productivity (please see our response above).

(8) For a given species richness, did the number of neighbors that were conspecific with the focal tree vary at all? If so, did focal trees, for a given neighborhood species richness, perform worse if there were more than if there were fewer conspecific neighbors?

Response: Yes, the number of conspecific neighbours varied within a given neighbourhood species richness (NSR) level as indicated by the negative correlation between NSR and CND ($r = -0.76$; note a perfect fit of 1 would indicate no variation of CND within a given NSR-level). We included this information in the main text (p15 L324-327) and the new Supplementary method. To test for conspecific neighbour density (CND) dependency on NSR effects, we used the interaction between NSR and CND in our candidate models. According to the AIC, we found no support for such interacting effects. Thus, the effect of NSR on individual tree growth did not depend on CND (see Supplementary Table 6). An alternative sequential testing approach (Type I ANOVA) gave similar results (Table R2).

Table R2: Effects of tree size (V), neighbourhood competition (NCI), conspecific neighbour density (CND) and neighbourhood species richness (NSR) on individual tree growth. *F*-values refer to Type I ANOVA with Satterthwaite approximation for degrees of freedom.

Fixed effect	d.f.	F -value	P -value	Fixed effect	d.f.	F -value	P -value
V (log)	3873.6	2755.1	< 0.001	V (log)	3873.6	2755.1	< 0.001
NCI (log+1)	1944.9	118.2	< 0.001	CND	273.6	1.8	0.183
CND	269.9	0.6	0.429	NCI (log+1)	2174.6	112.6	< 0.001
NSR	710.0	4.2	0.041	NSR	710.0	4.2	0.041
CND : NSR	897.7	0.2	0.661	CND : NSR	897.7	0.2	0.661

References

- Burnham, K. P. & Anderson, D. R. *Model selection and multimodel inference: a practical information – theoretic approach* (Springer, 2002).
- Burnham, K. P. & Anderson, D. R. & Huyvaert, K. P. AIC model selection and multimodel inference in behavioural ecology: some background, observations, and comparisons. *Behav. Ecol. Sociobiol.* **65**, 23–35 (2011).
- Coomes, D. A. & Allen, R. B. Effects of size, competition and altitude on tree growth. *J. Ecol.* **95**, 1084–1097 (2007).
- Fichtner, A. *et al.* From competition to facilitation: how tree species respond to neighbourhood diversity. *Ecol. Lett.* **20**, 892–900 (2017).
- Kröber, W. *et al.* Early subtropical forest growth is driven by community mean trait values and functional diversity rather than the abiotic environment. *Ecol. Evol.* **5**, 3541–3556 (2015).
- Lasky, J. R., Uriarte, M., Bouklic, V. K. & Chazdon, R. L. Trait-mediated assembly processes predict successional changes in community diversity of tropical forests. *Proc. Natl. Acad. Sci. USA* **111**, 5616–5621 (2014).
- Li, Y. *et al.* Site and neighborhood effects on growth of tree saplings in subtropical plantations (China). *For. Ecol. Manage.* **327**, 188–127 (2014).

- Li, Y., Kröber, W., Bruelheide, H., Härdtle, W. & von Oheimb, G. Crown and leaf traits as predictors of subtropical tree sapling growth rates. *J. Plant. Ecol.* **10**, 136-145 (2017)
- Paine, C. E. T. *et al.* Globally, functional traits are poor predictors of juvenile tree growth, and we do not know why. *J. Ecol.* **103**, 978-989 (2015).
- Valladares, F. & Niinemets, Ü. Shade tolerance, a key plant feature of complex nature and consequences. *Annu. Rev. Ecol. Evol. Syst.* **39**, 237–57 (2008).
- Schmid, B., Hector, A., Saha, P. & Loreau, M. Biodiversity effects and transgressive overyielding. *J. Plant Ecol.* **1**, 95-102 (2008).
- Zuur, A.F., Ieno, E.N., Walker, N.J., Saveliev, A.A. & Smith, G.M. *Mixed Effects Models and Extensions in Ecology with R.* (Springer, 2009).

Reviewer #2 (Remarks to the Author):

In their well written manuscript entitled “Neighbourhood interactions drive overyielding in mixed-species tree communities”, Fichtner and colleagues describe how local neighbourhood interactions between trees drive positive relationships between plot scale species richness and aboveground productivity.

The finding that neighbouring trees can promote the growth of a focal tree is not new, and has also been described in various other studies (although in the introduction section, the authors do suggest that there is a lack of published research on this). What distinguishes this work from other studies, is that as a second step, the authors investigated how these local diversity effects on individual tree growth scale up to the level of communities, and hence help explaining community level diversity-productivity relationships as observed in many forests. They conclude that “diversity-mediated interactions among local neighbours are more relevant for enhancing productivity in mixed-species forests than community tree species richness per se”. While this is of course a very interesting conclusion, I am not yet convinced that it is supported by the analyses, as this conclusion requires the testing of more ‘alternative hypotheses’.

Individual tree growth is statistically modelled as a response of the identity and richness of up to eight ‘neighbouring’ trees. However, what would happen if instead, individual tree growth is statistically modelled as a response of the identity and richness of more (within a larger radius), or fewer (e.g. only the closest) neighbouring trees? The authors could run various alternative models, differing in the spatial scale at which ‘neighbourhood’ is defined. (as a side note, I would like to note that it was not entirely clear to me how ‘neighbourhood’ was actually defined: as the tree individuals that are within a certain (e.g. 5m) radius of the focal tree? How was it decided whether 1, 2, 3 ... or 8 individuals were included?) The authors could then specifically look which type of neighbourhood is most relevant in driving tree growth. Only then, conclusions about the scale at which diversity matters most can be drawn.

Response: While we agree that testing for scale-dependence of biodiversity effects would be indeed a very interesting objective, this would be beyond the scope of our study. Instead, our main objective is to test the importance of local-scale interactions (i.e. immediate neighbourhood effects between local neighbouring trees) on community productivity rather than testing the importance of various spatial scales for biodiversity effects. This means that we were exclusively interested in the effects on the smallest spatial scale, where trees directly interacted with their direct, closest neighbours. Our focus on the local neighbourhood is

primarily based on the fact that species interactions at the local neighbourhood level are considered to be crucial to understand effects at the community level (Scherer-Lorenzen 2014), because plant interactions emerge at small spatial scales (Stoll & Weiner 2000). In fact, physical complementarity (for example such as canopy space use efficiency; Williams *et al.* 2017) and facilitative effects (such as microclimate amelioration; Fichtner *et al.* 2017) come about only by immediate neighbours. Thus, the mixing effects at the community scale is assumed to be – at least to a certain extent – the result of such aggregated local-scale variations in neighbourhood interactions (Potvin & Gotelli 2008; Potvin & Dutilleul 2009; Van de Peer *et al.* 2017, Williams *et al.* 2017). This explanation, which is directly linked to our hypotheses, is presented in the Introduction.

In this study, ‘local neighbourhood’ was defined as the total number of closest trees surrounding a focal tree. By using a planting scheme with a regular planting distance of 1.29 m, the local neighbourhood of a focal tree constitutes of maximum eight direct neighbours (a schematic diagram is given in Supplementary Figure 5). We rephrased several passages in the manuscript to more explicitly show that with neighbourhood we refer to the direct neighbouring trees (e.g. p6 L105-107, p15 L330 and abstract). Moreover, we thoroughly revised the abstract and conclusions by explicitly referring to the local neighbourhood scale and by avoiding statements on comparisons between different spatial scales (i.e. the scale at which diversity matters most; p3 L41-43, p11 L244-247).

I also have the comment regarding the presentation of results. The analyses consisted of two main parts: (1) the analysis of focal tree growth and (2) the analysis of community level tree productivity. While the results of the second part are clearly presented in the figures, the results of the first part are only described in the text and in the supplementary material, but not illustrated with figures in the main document. To really convince readers of the relevance of the findings, I think it would help to add at least one figure that illustrates the results of the first part of the study. For example, a graph showing the relationship between over yielding of focal tree growth and neighbourhood species richness (such as supplementary figure 1, but preferably also including the data points or at least the R², which would help assessing the strength of this relationship) would be extremely helpful.

Response: As suggested, we included a figure showing overyielding of focal tree growth and neighbourhood species richness for varying tree sizes and competition indices in the main text (new Fig. 1). The figure illustrates the highly significant 3-way interaction between tree size, neighbourhood competition (NCI) and species richness of our best-fitting neighbourhood

model. As the regression lines represent the predicted response for a specific tree size and NCI combination, it is not possible to include raw data. However, we included a sentence in the figure caption, reporting the r^2 value (p29 L639-640).

Regarding Fig. S3: I found it slightly concerning the standardised effect of neighbourhood AWP is above one. While this is theoretically possible, standardised effects with absolute values greater than one are often a sign of multicollinearity. I realise that the authors tested for a multicollinearity and reported low VIF values, but I do wonder whether this extremely high standardised effects might have been caused by a strong relationship between neighbourhood AWP and one of the random effects? I urge the authors to have a serious look at this, as such extremely standardised effects may be a sign of a flawed model.

Response: We believe that these concerns are based on a misunderstanding, probably arising from an unclear description of our methodological approach. In our study, we standardised all predictors, but not the response variable (Schielzeth 2010). Thus, it is mathematically possible to have estimated coefficients outside of -1 to 1, depending on the scale of the response variable. In the previous version of the manuscript we log-transformed both the response variable and predictors by using the natural logarithm (ln). This resulted in an estimated coefficient for neighbourhood AWP of 1.072. To avoid confusion, we used the logarithm with base 10 (log10) in the revised version of the manuscript, which resulted in an estimated coefficient for neighbourhood AWP of 0.457 (see new Supplementary Figure 4). To avoid misunderstandings and to clarify the standardisation approach, we revised the method section accordingly (p15 L342-343, p17 L378, p19 L433-434).

Additional comments

L 30-31: meta-analyses such as Cardinale et al, 2011, Am. J. Bot. suggest something else, namely that biodiversity effects are largest at large scales, and suggest that this may be due to large scale spatial heterogeneity in e.g. resources

Response: We rephrased this sentence and focused explicitly on tree communities and recent findings from tree biodiversity experiments (p3 L30-31).

We agree that the synthesis of Cardinale et al. from biodiversity communities in 30 biomes suggests that biodiversity effects increase with the spatial scale considered (“*The assumption is that the larger spatial scales and greater temporal fluctuations that are typical of natural systems incorporate more heterogeneity and more niche opportunities for species to exploit resources than are available.*”) However, this conclusion is largely based on grasslands,

because forests are underrepresented in this study (as the authors state that “*tropical and temperate forests*” are “*systems for which there is a relatively small amount of data*”). In contrast, recent results from tree biodiversity experiments suggest that for both tropical and temperate forests the local neighbourhood level should be a main driver for positive biodiversity-productivity relationships at the community level (Gotelli & Potvin 2008; Potvin *et al.* 2009; Ratcliffe *et al.* 2015; Williams *et al.* 2017; van de Peer *et al.* 2017). We therefore assume that our revised sentence is now supported by these findings.

L 43-48: doesn't the sentence of line 38-40 suggest that neighbourhood and community richness should both be preserved, as they strengthen each other's effects? In that case, the conclusion needs to be rephrased.

Response: We agree and rephrased the conclusion section accordingly (p3 L41-43).

L 54-56: there is actually quite a long list of studies that investigated (taxonomic / functional / phylogenetic) diversity-productivity/growth relationships at the neighbourhood scale: e.g. Potvin & Gotelli, 2008, Uriarte *et al.*, 2010; Lubbe *et al.*, 2015, *Front. In Plant Sci.*; Kunstler *et al.*, 2016, *Nature; Ecol. Lett.*; del Rio *et al.*, 2017, *J. of Ecol.*; Dillen *et al.*, 2017, *Forest Ecol. & Managm.* and the authors own, previous work (Fichtner *et al.*, 2017, *Ecol. Lett.*), to name just some articles (I am sure there are more).

Response: We thoroughly revised the first paragraph by avoiding the statement that “most” studies investigated diversity effects at the community scale (p4 L48-56). Instead, we cited experimental studies that investigated biodiversity-productivity relationships in tree communities at both the neighbourhood and community scale. While these studies found positive biodiversity effects at both scales, the contribution of neighbourhood interactions to community productivity (i.e. the quantification of such local-scale effects), however, is still largely unknown.

L 61: niche differentiation is a bit vague, as niches can be very broadly defined. So I would suggest ‘resource partitioning’ as an alternative

Response: Done (p4 L60).

L 65-66: it should be added that these well performing species also need to dominate in order to get positive selection effects.

Response: We rephrased this sentence and added “dominant” (p4 L65).

L 94-98: rephrase the sentence, as there is a lot of redundancy within it. Currently, it is basically stated twice what are your objectives

Response: We revised this paragraph to avoid redundancy (p5 L89-95).

L 99: so you basically expect a positive interaction effect between neighbourhood and plot-scale richness? Why?

Response: We gave more detailed information that supports this hypothesis in the previous paragraph (p5 L77-88).

L115: add 'and compare' after 'obtain'

Response: Done (p6 L112).

L 137: what is meant with 'consistently'?

Response: We rephrased this sentence and deleted "consistently" (p7 L135-136).

Fig. 2: the panels of this figure are too small, making it almost impossible to read. I suggest to rearrange it into a 2x2 panel figure.

Response: As suggested, we re-arranged the figure in 2 x 2 panels (now Figure 3)

L 187: local -> neighbourhood

Response: We added "neighbourhood" (p8 L166).

L 203-205: quite likely, aboveground size is highly related to belowground size. In that case, results could be just as well be driven by belowground competition for e.g. nutrients or water, especially in deeper soil layers.

Response: We agree and therefore added a sentence addressing this issue (p9 184-187). Moreover, we toned down our previous statement by saying "relative competition intensity via (light) resource depletion" (p9 187-190).

L 211: add 'a' after 'as'

Response: Done (p9 L192).

L 214-217: please rephrase, this sentence is hard to understand. Also, given the overall negative NCI effects, is it really likely that facilitation plays a key role in this study? Or is competition simply weakened by species richness?

Response: Done (p9 L195-197).

We agree that facilitation might be of minor importance, probably due to our across-species response. In a previous study (Fichtner *et al.* 2017), using data from the same study site, we found that competitive reduction and facilitation strongly depend on focal tree's functional traits and that facilitation was highest for large-sized trees. Facilitation is generally defined as the standardized ratio between the performances of a target plant measured with and without neighbours (see Armas *et al.* 2004; Díaz-Sierra *et al.* 2017). Positive values of this ratio indicate facilitation, while negative values refer to competition (for a conceptual overview in relation to neighbourhood species richness see Figure 1 in Fichtner *et al.* 2017). Thus, the fact that for large-sized trees net diversity effects were higher at high compared to low values of neighbourhood competition (NCI) indices suggests that facilitation, rather than competitive reduction drive tree interactions at the local neighbourhood scale.

L 239-243: unclear sentence, please rephrase

Response: Done (p10 L 219-221).

L 276: site -> sites

Response: Done. Thanks!

L 313: aren't their more than 2 measurements per tree, as measurements were done each year from 2011-2016?

Response: Yes, trees were measured every year. To avoid confusion, we revised this sentence by explicitly saying that we analysed a 5-year (2011-2016) study period. Thus, tree data must be available for 2011 and 2016 to calculate growth rates (p13 L286-289).

L 325: what is 'form factor'?

Response: We added an explanation for the form factor used (p14 L302-303). A form factor is a reduction factor that reduces the theoretical volume of a cylinder to tree volume (related to a tree's slenderness; Pretzsch 2009). This is a common approach for the calculation of tree volume in e.g. forestry.

L 333-337: as a sensitivity analysis, I would like to see how this choice affected the results

Response: We assume no serious effect by excluding negative growth rates. In total, we excluded just 68 trees (site A) and 62 trees (site B) from a total sample size of 3962 trees (1.7% of the site A data) and 3018 trees (2.1% of the site B data), respectively. It was provable during the inventories that negative growth rates (maximum negative growth rates

ranged between $-583 \text{ cm}^3 / \text{year}$ (site A) and $-55.6 \text{ cm}^3 / \text{year}$ (site B)) clearly resulted from tree damage. Including such data would result in spurious correlations.

Moreover, we have to apply a log-transformation to the response variable to meet model assumptions, as positive growth rates ranged between 0.01 and $62510 \text{ cm}^3 / \text{year}$ for site A, and between 0.004 and $29740 \text{ cm}^3 / \text{year}$ for site B. Including negative growth rates would therefore require to add a constant of 584 and 56, respectively. This would also result in biased estimates, particularly for small growth rates. To avoid biased parameter estimates, we therefore decided to remove these data prior to analysis.

L 344-347: as a sensitivity analysis, it would be good to also quantify NCI based on the biomass of all individuals, including those that are smaller than the focal tree, as it is possible that smaller individuals also have competitive effects.

Response: We run a sensitivity analysis and compared models with neighbourhood competition indices (NCI) based on the total basal area of neighbours larger than the focal tree (BAL) or NCI based on the total basal area of all neighbours (BA). Both competition indices were qualitatively the same (Table R3, also see new Supplementary Table 9). However, the best-fitting neighbourhood model using NCI based on size-asymmetric competition (BAL) provided a significant better fit to the data compared to the size-symmetric NCI (BA; $\Delta\text{AIC} = 184.3$, $P < 0.001$), and was therefore preferred. Moreover, the Akaike weights indicated that the model including size-asymmetric NCI effects has a relative likelihood being the best fitting model of 100% and lower mean absolute error and root mean squared error (Table R3).

We included this information in a new section “Sensitivity analysis” in the Supplementary Methods.

Table R3. Standardised regression coefficients estimates for the best-fitting neighbourhood model (site A) using NCI based on total basal area of larger neighbours (BAL) and total basal area of all neighbours (BA). Akaike Information Criterion (AIC) and error statistics (MAE: mean absolute error and RSME: root mean squared error) indicated a better fit for the model using NCI based on BAL.

	NCI based on BAL		NCI based on BA	
	Estimate	P-value	Estimate	P-value
Fixed effects				
Intercept	1.941	< 0.001	1.385	< 0.001
Initial focal tree volume (V, log)	0.536	< 0.001	0.711	< 0.001
Neighbourhood competition index (NCI, log)	-0.273	< 0.001	-0.048	0.389
Neighbourhood tree species richness (NSR)	0.229	0.002	0.239	< 0.001
V * NCI	0.064	< 0.001	0.011	0.694
V * NSR	-0.066	0.011	-0.055	0.052
NCI * NSR	-0.069	0.010	-0.117	< 0.001
V * NCI * NSR	0.026	0.007	0.032	0.025
Random effects				
SD (plot)	0.280		0.27	
SD (species identity)	0.440		0.47	
SD (neighbourhood species composition)	0.093		0.11	
SD (neighbour density)	0.046		0.07	
SD (residuals)	0.497		0.51	
Model fit				
AIC	6353.3		6537.6	
r^2_m	0.48		0.48	
r^2_c	0.76		0.76	
MAE	0.54		0.55	
RMSE	0.69		0.71	

L 356-359: it would be good to actually test whether these relationships are indeed linear, which could be done by also running an alternative model with additionally a quadratic term, and to investigate whether the simpler, linear model is more parsimonious than the model also including a quadratic term.

Response: In this study, we linearized the relationship between absolute growth rate (AGR) and competition by log-transforming both AGR and tree size and neighbourhood competition index (NCI). The log-log transformation assumes a power-law function on the original scale (Harrel 2001). Moreover, linearization reduces the effect of outliers (especially in covariates) and stabilises the variance (reduction in heteroscedasticity), and thus leads to unbiased parameter estimates (Zuur *et al.* 2010). It is also important to note that most tree diversity studies linearized relationships by applying log-transformation for both the dependent and independent variable, and thus, in order to account for potential non-linear covariate effects (as examples for tropical and sub-tropical tree species: Rüger *et al.* 2011, 2012; Chen *et al.* 2016; for temperate and boreal tree species: Jucker *et al.* 2014, 2016; Ratcliffe *et al.* 2015;

Forrester *et al.* 2016; Chamagne *et al.* 2017; for tree species on a global scale: Kunstler *et al.* 2016). We therefore consider this an appropriate approach which also facilitates comparisons among tree diversity studies.

Nevertheless, in order to validate our linearization approach (linear log-log relationship), we furthermore examined whether there is a non-linear pattern in the residuals. We first plotted the residuals of our minimum-adequate model versus initial tree size (V) and neighbourhood competition index (NCI). The plots suggest that there is no non-linear pattern. However, visual diagnostics might be difficult to validate. Therefore, we modelled the residuals of our minimum-adequate models as a smoothing function of V and NCI, following Zuur *et al.* (2009) and Zuur & Ieno (2016a). If the smoother is not significant and/or only explains a small proportion of the variation in the residuals then there is no support for non-linearity (Zuur & Ieno 2016a,b). For both V and NCI, the models indicated non-significant smoother and a very low explained proportion of the variation in the residuals by non-linear V and NCI effects (Figure R2). Thus, there is no indication of a non-linear residual pattern caused by V or NCI and we assume unbiased parameter estimates and standard errors of our neighbourhood models.

Figure R2: Pearson residuals of the minimum-adequate linear-mixed effects model plotted versus initial tree volume (V) and neighbourhood competition index (NCI). The smoothers (red line) and corresponding r^2 - and P -values were derived from generalised additive models with a thin plate regression spline smoother. The shaded areas represent the 95% confidence interval range.

L 389-390: doesn't this contradict L383-384? Please clarify

Response: In this study, we used the Akaike Information Criterion (AIC) as model selection strategy, because we were primarily interested in determining systematically which predictors and which predictor combination would result in a most parsimonious model, without having a priori expectations on the importance of predictors (i.e. sequential order of predictors). Therefore, we used a set of various candidate models. These models are nested. As models with nested fixed effects (but with the same random structure) cannot be compared with the restricted maximum likelihood (REML) method, we have to use the maximum likelihood (ML) method for the selection of the optimal fixed-effects structure (e.g. Crawley 2007; Zuur *et al.* 2009).

L 391: sites -> site; add “strong” before “correlation”

Response: Done. Moreover, we added “critical” before “correlation” and hope that is in accordance with the reviewer's suggestion (p17 L379).

L 438: given that all plots were very close to each other, how accurate are the GPS measurements in describing local scale variation in altitude, northness and slope?

Response: A digital elevation model (DEM) with a cell size of 5×5 m was interpolated from elevation measurements with differential global positioning system (accuracy: 5-10 cm) using the ordinary kriging algorithm. The overall quality of the DEM was high, with an explained variance of 98% and a root mean square error (RMSE) of 1.9 m (10-fold cross validation) in an elevation range of 112 m (Li *et al.* 2014). We included this information in the main text (p19 L427-431).

Supplementary material Fig. S1, left panel: how can the diversity effect be above 0 when species richness is 1?

Response: The variable neighbourhood species richness (NSR) accounts for the number of different tree species in the local neighbourhood of a focal tree in relation to the species identity of the focal tree. NSR was calculated as the total number of closest heterospecific neighbour species ($\sum_{j \neq i} N_j$, where N is the recorded species number; p15 322-323, L 331). Thus, the minimum value of NSR is 0, meaning that all neighbours have the same species identity as the focal tree. Figure S1 (now Fig. 1) displays the relative change in predicted growth rate (G) of a focal tree growing in conspecific (NSR=0) compared to heterospecific neighbourhoods (NSR=1,...,7) - please also see Supplementary Methods “Quantifying changes in tree growth along tree species richness gradients”. Thus, values at NSR=1 refer to

the growth rate relative to NSR=0. To enhance clarity, we added “(NSR=0)” and “(NSR=1,...,7)” in the Supplementary Methods.

Also, the over yielding effects seem relatively small (up to ~10%). How come that the community level diversity effects are much larger?

Response: The magnitude of overyielding Fig S1 (now Fig. 1) refers to the average overyielding of an individual tree, while the community level diversity effects are considered as an aggregate effect of individual tree responses. Thus, the community diversity effect is the net effect of the individual tree overyielding. However, it is important to note that we found a strong context-dependency of the magnitude of diversity effects at the individual tree level (3-way interaction between focal tree size and neighbourhood competition and tree species richness). Moreover, we showed that factors other than neighbourhood interactions drive overyielding at the community scale.

References

- Armas, C., Ordiales, R. & Pugnaire, I. Measuring plant interactions: a new comparative index. *Ecology* **85**, 2682–2686 (2004).
- Chamagne, J. *et al.* Forest diversity promotes individual tree growth in central European forest stands. *J. Appl. Ecol.* **54**, 71–79 (2017).
- Chen, Y. *et al.* Positive effects of neighborhood complementarity on tree growth in a Neotropical forest. *Ecology* **97**, 776–785 (2016).
- Crawley, M. J. *The R Book* (Wiley & Sons, 2007).
- Díaz-Sierra *et al.* A new family of standardized and symmetric indices for measuring the intensity and importance of plant neighbour effects. *Methods Ecol. Evol.* **8**, 580–591 (2017).
- Fichtner, A. *et al.* From competition to facilitation: how tree species respond to neighbourhood diversity. *Ecol. Lett.* **20**, 892–900 (2017).
- Forrester, D.I., Benneter, A., Bouriaud, O. & Bausch, J. Diversity and competition influence tree allometric relationships – developing functions for mixed-species forests. *J. Ecol.* **105**, 761–774 (2017).

- Harrel, F. E. *Regression Modelling Strategies* (Springer, 2001).
- Jucker, T. *et al.* Competition for light and water play contrasting roles in driving diversity–productivity relationships in Iberian forests. *J. Ecol.* **102**, 1202–1213 (2014).
- Jucker, T. *et al.* Climate modulates the effects of tree diversity on forest productivity. *J. Ecol.*, **104**, 388–398 (2016).
- Li, Y. *et al.* Site and neighborhood effects on growth of tree saplings in subtropical plantations (China). *For. Ecol. Manage.* **327**, 188–127 (2014).
- Kunstler, G. *et al.* Plant functional traits have globally consistent effects on competition. *Nature* **529**, 204–207 (2016).
- Potvin, C. & Gotelli, N. J. Biodiversity enhances individual performance but does not affect survivorship in tropical trees. *Ecol. Lett.* **11**, 217–223 (2008).
- Potvin, C. & Dutilleul, P. Neighborhood effects and size-asymmetric competition in a tree plantation varying in diversity. *Ecology* **90**, 321–327 (2009).
- Pretzsch, H. *Forests dynamics, growth and yield.* (Springer, 2009).
- Ratcliffe, S., Holzwarth, F., Nadrowski, K., Levick, S. & Wirth, C. Tree neighbourhood matters – Tree species composition drives diversity–productivity patterns in a near-natural beech forest. *For. Ecol. Manage.* **335**, 225–234 (2015).
- Rüger, N., Berger, U., Hubbell, S.P., Vieilledent, G. & Condit, R. Growth strategies of tropical tree species: disentangling light and size effects. *PLoS ONE* **6**, e25330 (2011).
- Rüger, N., Wirth, C., Wright, S.J. & Condit, R. Functional traits explain light and size response of growth rates in tropical tree species. *Ecology* **93**, 2626–2636 (2012).
- Scherer-Lorenzen, M. The functional role of biodiversity in the context of global change. In: *Forests and global change* (eds Coomes, D.A., Burslem, D.F.R.P. & Simonson, W.D.), 195–237 (Cambridge University Press, 2014).
- Schielzeth, H. Simple means to improve the interpretability of regression coefficients: Interpretation of regression coefficients. *Methods Ecol. Evol.* **1**, 103–113 (2010).
- Stoll, P. & Weiner, J. A. A neighborhood view of interactions among individual plants. In *The geometry of ecological interactions: Simplifying spatial complexity*, (eds Dieckmann, U., Law, R. & Metz, J. A. J.), 11–27 (Cambridge University Press, 2000).

- Van de Peer, T., Verheyen, C., Ponette, Q., Setiawan, N. N. & Muys, B. Overyielding in young tree plantations is driven by local complementarity and selection effects related to shade tolerance. *J. Ecol.*, Early view, DOI: 10.1111/1365-2745.12839 (2017).
- Williams, L. J., Paquette, A., Cavender-Bares, J., Messier, C. & Reich, P. B. Spatial complementarity in tree crowns explains overyielding in species mixtures. *Nat. Ecol. Evol.* **1**, 0063 (2017).
- Zuur, A. F., Ieno, E. N., Walker, N. J., Saveliev, A. A. & Smith, G. M. *Mixed Effects Models and Extensions in Ecology with R* (Springer, 2009)
- Zuur, A. F., Ieno, E. N. & Elphick, C. S. A protocol for data exploration to avoid common statistical problems. *Methods Ecol. Evol.* **1**, 3–14 (2010).
- Zuur, A. F. & Ieno, E. N. A protocol for conducting and presenting results of regression-type analyses. *Methods Ecol. Evol.* **7**, 636–645 (2016a).
- Zuur, A. F. & Ieno, E. N. *Beginner's guide to zero-inflated models with R*. Highland Statistics Ltd., UK, 2016b).

REVIEWERS' COMMENTS:

Reviewer #2 (Remarks to the Author):

I have read the revised version of the study by Fichter and colleagues entitled "Neighbourhood interactions drive overyielding in mixed-species tree communities". Previously, I had some concerns about the framing of the article, the framing of the results and about the statistical analyses, as well as about various smaller aspects. In general, I think that the authors reacted in a constructive way to this feedback, by revising and clarifying various parts of the manuscript, and their responses to the suggestions were generally strong. Therefore, this time I do not have any major concerns, although I do have several smaller suggestions for further improving this already excellent study, which are outlined below.

L 30: "biodiversity effects": I would change this into "plant/tree interactions", as otherwise the sentence is a bit circular (biodiversity effects causing biodiversity-productivity relationships)

L 43: "in the face of global biodiversity loss": global and local diversity loss are not necessarily related, so I would omit this.

L 78: "species" -> "individuals", as individuals from the same species also interact

L83-86: "For example, simulation models revealed that neighbourhood interactions can induce positive BPRs in tree communities, but their role in regulating BPRs at the community scale is still poorly understood". The second part of the sentence (after second comma) seems to contradict the first part, or am I overlooking something?

L 212: why do aboveground-belowground interactions primarily operate at larger spatial scales? I also have the same question for the negative density-dependency of herbivory/pathogens: I would say that one could also argue that these processes are especially important at local scales.

L 237-241: neighbourhood richness effects are only one component of neighbourhood interactions (another important one being the NCI). So one could alternatively argue that NCI (or any other neighbourhood factor) becomes more important with a higher community richness.

L 346-347: this conclusion seems preliminary. I can imagine that trees with larger DBH also have larger root systems, as the growth of most plant organs presumably scale with diameter increment. So one could argue just as well that NCI is a proxy for belowground competition. In reality, we don't really know which one is most important, so I would suggest to simply remove this sentence.

Line 358: is a truly the overall mean growth rate? I would rather call it the intercept (which often deviated from the mean, although the authors can easily check this).

L 370: "more" -> "most"

L 471: "is the is the" -> "is the"

Fig. 3: this figure consists of 4 panels, which represent different community-diversity levels. However, since AWP depends on both community and neighbor diversity levels, it would be insightful to have an additional panel for the complete dataset.

Supplementary Table 2: previously, I had a concern because of a standardized regression coefficient exceeding 1. As the authors replied, this was due to a misunderstanding: only predictor variables were standardized, whereas the response is not, so that coefficients below -1 or above +1 are possible.

While this is a satisfactory response, I would recommend one more thing: which is to not call these coefficients "standardized" in the caption (line 108), as the "standardized regression coefficients" suggest that both predictor AND response are standardized.

Supplementary table 3 & 4: same comment as previous.

[Editor's Notes: As Reviewer #1 was unavailable to comment on the most recent version of the manuscript, Reviewer #2 agreed to comment in their place. These comments appear below]

One of the comments on the previous version of this manuscript was that the effect of local species richness on focal tree growth that the authors observed, was in fact spurious, and driven by the covarying factor of local conspecific density. It was recommended to test for this, by accounting for conspecific density in the statistical models, and by testing whether the effects of local species richness remained significant and strong when this other factor was taken into account.

Although a sequential model selection was advised, in which first conspecific neighbourhood density was included, and only later local species richness, the authors chose for a slightly different approach, in which they tested all possible types of models and investigated which one most parsimoniously explained local tree growth. In addition, the authors used different types of significance tests to investigate whether models including local richness (which did indeed correlate rather strongly negatively with local conspecific density) were more significant than models including local conspecific density. While they did not follow 100% of the reviewer's recommendations, their response did adequately address the author's question whether local richness effects were in fact driven by conspecific density, with, irrespective of the approach, evidence that local richness was more important than conspecific density. I am therefore confident that the original conclusions were not biased by variation in local conspecific density.

Another main concern included the possibility of spatial autocorrelation influencing the results. However, in their response the authors convincingly showed that this was unlikely a major problem. In addition, there was some discussion on why growth rate variation was higher in monocultures than in diverse communities. While this discussion is not hugely important for this study (and I don't expect the authors to make additional changes), I found the author's explanation "because for a given species richness level extreme values are more extreme the larger the sample size is (Schmid et al. 2008)" a bit vague. Perhaps what they meant, is that in diverse communities, species with high and low growth rates tend to coexist, and that the community-wide growth therefore tends to be an average of these species specific growth rates. Thus, extremely high or low rates should be rare, something described in more mechanistic detail by Loreau, 1998, PNAS and also for other functions than biomass alone by van der Plas et al. 2016 and Gamfeldt & Roger, 2017, Nature Ecology and Evolution. Again, I don't think this discussion is very important for the manuscript (so no changes needed), but I just thought this might be interesting mentioning.

In addition, there were various other, smaller comments, which I think the authors addressed adequately. As a result, I believe the latest version of this manuscript is a clear improvement compared to the earlier version.

We are grateful to the reviewer for further constructive comments to improve our manuscript. We have considered all the suggestions made by the reviewer and have revised the manuscript accordingly.

Please find below a detailed point-by-point response and a description of the changes (each referring to line numbers in the revised manuscript “Fichtner et al. Main text_R2_clean edit”) related to the comments of the reviewer. Note that we additionally fixed a typo regarding the site coordinates (L256).

Reviewers' comments:

Reviewer 2:

I have read the revised version of the study by Fichter and colleagues entitled “Neighbourhood interactions drive overyielding in mixed-species tree communities”. Previously, I had some concerns about the framing of the article, the framing of the results and about the statistical analyses, as well as about various smaller aspects. In general, I think that the authors reacted in a constructive way to this feedback, by revising and clarifying various parts of the manuscript, and their responses to the suggestions were generally strong. Therefore, this time I do not have any major concerns, although I do have several smaller suggestions for further improving this already excellent study, which are outlined below.

L 30: “biodiversity effects”: I would change this into “plant/tree interactions”, as otherwise the sentence is a bit circular (biodiversity effects causing biodiversity-productivity relationships)

Response: We changed “biodiversity effects” into “plant interactions”.

L 43: “in the face of global biodiversity loss”: global and local diversity loss are not necessarily related, so I would omit this.

Response: Omitted as recommended. Moreover, we deleted two times “tree” (L32, L37) to meet word count requirements.

L 78: “species” -> “individuals”, as individuals from the same species also interact

Response: Revised as recommended.

L83-86: “For example, simulation models revealed that neighbourhood interactions can induce positive BPRs in tree communities, but their role in regulating BPRs at the community scale is still poorly understood”. The second part of the sentence (after second comma) seems to contradict the first part, or am I overlooking something?

Response: We rephrased this sentence to more explicitly state that the contribution of local neighbourhood interactions to biodiversity effects at the community scale remains largely unknown, although a simulation study provided first insights that neighbourhood complementarity can lead to positive biodiversity effects at the community scale (L81-84).

L 212: why do aboveground-belowground interactions primarily operate at larger spatial scales? I also have the same question for the negative density-dependency of herbivory/pathogens: I would say that one could also argue that these processes are especially important at local scales.

Response: We rephrased this sentence to stress the importance of these processes at the local neighbourhood scale (L208-212).

L 237-241: neighbourhood richness effects are only one component of neighbourhood interactions (another important one being the NCI). So one could alternatively argue that NCI (or any other neighbourhood factor) becomes more important with a higher community richness.

Response: We deleted “diversity-mediated” and used the more neutral term “neighbourhood interactions” to acknowledge factors others than neighbourhood species richness (e.g. neighbourhood competition) that may co-determine tree interactions at the local neighbourhood scale (L236).

L 346-347: this conclusion seems preliminary. I can imagine that trees with larger DBH also have larger root systems, as the growth of most plant organs presumably scale with diameter increment. So one could argue just as well that NCI is a proxy for belowground competition. In reality, we don’t really know which one is most important, so I would suggest to simply remove this sentence.

Response: Done as requested.

Line 358: is a truly the overall mean growth rate? I would rather call it the intercept (which often deviated from the mean, although the authors can easily check this).

Response: We changed “overall mean growth rate” into “intercept”.

L 370: “more” -> “most”

Response: Thank you. Done.

L 471: “is the is the” -> “is the”

Response: Thank you. Done.

Fig. 3: this figure consists of 4 panels, which represent different community-diversity levels. However, since AWP depends on both community and neighbor diversity levels, it would be insightful to have an additional panel for the complete dataset.

Response: We revised this Figure and added an additional panel showing the relationship across all monocultures and species mixtures (i.e. the complete dataset).

Supplementary Table 2: previously, I had a concern because of a standardized regression coefficient exceeding 1. As the authors replied, this was due to a misunderstanding: only predictor variables were standardized, whereas the response is not, so that coefficients below -1 or above +1 are possible. While this is a satisfactory response, I would recommend one more thing: which is to not call these coefficients “standardized” in the caption (line 108), as the “standardized regression coefficients” suggest that both predictor AND response are standardized.

Supplementary table 3 & 4: same comment as previous.

Response: Done as requested. We additionally deleted “standardised” in Figure S4 and the respective caption.